# Kronecker Factorization Improves Efficiency and Interpretability of Sparse Autoencoders

## Abstract

Sparse Autoencoders (SAEs) have demonstrated significant promise in interpreting the hidden states of language models by decomposing them into interpretable latent directions. However, training and interpreting SAEs at scale remains challenging, especially when large dictionary sizes are used. While decoders can leverage sparse-aware kernels for efficiency, encoders still require computationally intensive linear operations with large output dimensions. To address this, we propose Kron-SAE – a novel architecture that factorizes the latent representation via Kronecker product decomposition, drastically reducing memory and computational overhead. Furthermore, we introduce mAND, a differentiable activation function approximating the binary AND operation, which improves interpretability and performance in our factorized framework.

## 1 Introduction

Interpreting large language models and their embeddings in particular remains a central challenge for transparency and controllability in AI systems (Elhage et al., 2023; Heap et al., 2025). Sparse autoencoders (SAEs) have emerged as powerful tools for uncovering human-interpretable features within neural activations by enforcing activation sparsity to induce discrete-style dictionaries (Elhage et al., 2023; Gao et al., 2025; Cunningham et al., 2024). These dictionaries facilitate circuit-level semantic analysis (Marks et al., 2025) and concept discovery, enabling fine-grained probing of model internals (Elhage et al., 2023; Cunningham et al., 2024).

However, naively scaling SAEs to the widths demanded by modern transformers leads to prohibitive compute costs, limiting their applicability to large-scale interpretation experiments. *Gated* SAEs (Rajamanoharan et al., 2024a) address this by learning continuous sparsity masks via lightweight gating networks, and *Switch* SAEs (Mudide et al., 2025) leverage conditional computation by routing activations among smaller expert SAEs, reducing computation by activating only a subset of experts per input. Gao et al. (2025) propose an acceleration of TopK SAE that utilizes an optimized kernel based on efficient sparse–dense matrix multiplication. Encoder remains unoptimized: it still performs a dense projection into the full dictionary, incurring high computational cost and limiting scalability.

Another limitation of SAEs is the absence of structure within learned latents in its classical design. While Mudide et al. (2025) address this via expert subnetworks, Bussmann et al. (2025) imposes feature hierarchy via nested dictionaries structure and improves the interpretability of SAE latents.

In this paper we address both these directions and introduce **KronSAE**, an encoder that is applicable to many existing SAE architectures. By decomposing the latent space into head-wise Kronecker factors and including differentiable logical AND-like gating mechanism, KronSAE reduces both parameters and compute overhead while preserving reconstruction fidelity and improves interpretability.

This work makes three primary contributions:

1. We identify the encoder projection as one of the principal scalability bottlenecks in sparse autoencoders, demonstrating that targeted encoder optimizations can significantly improve computational performance while maintaining reconstruction quality.

2. We propose **KronSAE**, a Kronecker-factorised sparse autoencoder equipped with the novel mAND activation function. The design reduces encoder cost and is compatible with existing sparse decoder kernels, loss function designs and mixture of experts architecture.

3. We show on multiple language models that KronSAE decreases feature absorption and yields more interpretable latents under fixed compute.

## 2 RELATED WORK

**Sparse Autoencoders.** Early work demonstrated that SAEs can uncover human-interpretable directions in deep models (Cunningham et al., 2024; Templeton et al., 2024), but extending them to the large latent sizes ($F \gg d$) for modern language models with large hidden dimension $d$ is expensive: each forward pass still requires a dense $\mathcal{O}(Fd)$ encoder projection. Most prior optimization efforts focus on the decoder side: for example, Gao et al. (2025) introduce a fused sparse–dense TOPK kernel that reduces wall-clock time and memory traffic, while Rajamanoharan et al. (2024a) decouple activation selection from magnitude prediction to improve the $\ell_0$–MSE trade-off. Separately, Rajamanoharan et al. (2024b) propose JUMPRELU, a nonlinearity designed to mitigate shrinkage of large activations in sparse decoders.

**Conditional Computation.** These schemes avoid instantiating a full dictionary per token by routing inputs to a subset of expert SAEs via a lightweight gating network (Mudide et al., 2025) by employing the Mixture-of-Experts ideas (Shazeer et al., 2017), but still incur a dense per-expert encoder projection, leaving the encoder as the primary bottleneck.

**Factorizations and Logical Activation Functions.** Tensor product representations (Smolensky, 1990) have been utilized to represent the compositional structures in dense embeddings, closely resembling our idea of AND-like compositions, and recent studies have been extended TPR to study the transformer hidden states and in-context learning (Soulos et al., 2020; Smolensky et al., 2024). Separately, tensor-factorization methods have been used to compress large weight matrices in language models (Edalati et al., 2021; Wang et al., 2023), and structured matrices have been utilized to improve training efficiency and might be used to impose the inductive bias (Dao et al., 2022). In parallel, differentiable logic activations were introduced to approximate Boolean operators in a smooth manner (Lowe et al., 2021). Our method synthesizes these lines of work: we embed a differentiable AND-like gate into a Kronecker-factorized efficient encoder to build compositional features while preserving end-to-end differentiability.

## 3 METHOD

**Preliminaries.** Let $\mathbf{x} \in \mathbb{R}^d$ denote an activation vector drawn from a pretrained transformer. A conventional TopK SAE (Gao et al., 2025) produces a reconstruction $\hat{\mathbf{x}}$ of $\mathbf{x}$ via

$$\mathbf{f} = \text{TopK}\big(W_{\text{enc}}\mathbf{x} + \mathbf{b_{enc}}\big), \qquad \hat{\mathbf{x}} = W_{\text{dec}}\mathbf{f} + \mathbf{b_{dec}}, \tag{1}$$

where $W_{\text{enc}} \in \mathbb{R}^{F \times d}$ and $W_{\text{dec}} \in \mathbb{R}^{d \times F}$ are dense matrices and $\mathbf{f} \in \mathbb{R}^F$ is a sparse vector retaining only the $K$ largest activations. The encoder cost therefore scales as $\mathcal{O}(Fd)$ per token.

**KronSAE.** Our method *reduces the encoder's computational cost* while also *enforcing compositional structure* of the latents. We decompose the latent space into $h$ independent heads, and each head $k$ is parameterised by the composition of two thin matrices $P^k \in \mathbb{R}^{m \times d}$ (composition base) and $Q^k \in \mathbb{R}^{n \times d}$ (composition extension), with dimensions $m \le n \ll d$ and $F = h\,m\,n$. The *pre-latents*

$$\mathbf{p}^k = \text{ReLU}(\mathbf{u}^k) \qquad \text{and} \qquad \mathbf{q}^k = \text{ReLU}(\mathbf{v}^k), \tag{2}$$

with $\mathbf{u}^k = P^k\mathbf{x}$ and $\mathbf{v}^k = Q^k\mathbf{x}$ acting as the elements from which compositional features would be built, are combined through an element-wise interaction kernel independently in each head:

$$z_{i,j}^k := \text{mAND}(u_i^k, v_j^k) := \begin{cases} \sqrt{u_i^k\,v_j^k}, & u_i^k > 0 \text{ and } v_j^k > 0, \\ 0, & \text{otherwise}, \end{cases} \tag{3}$$

where $\mathbf{z}^k \in \mathbb{R}^{m \times n}$; it is then flattened in a row-major order to a vector equivalent to the element-wise square root of the Kronecker product of $\mathbf{p}^k$ and $\mathbf{q}^k$. Assuming that $\text{vec}(\cdot)$ is in row-order, we get

$$\text{vec}(\mathbf{z}^k) = \sqrt{\text{vec}\big(\mathbf{p}^k(\mathbf{q}^k)^\top\big)} = \sqrt{\mathbf{p}^k \otimes \mathbf{q}^k} \tag{4}$$

Concatenating heads and applying TopK yields *post-latents* $\mathbf{f} \in \mathbb{R}^F$. By ensuring active $z_{i,j}$ only when $p_i > 0$ and $q_j > 0$ we directly force the AND-like interaction: let $\mathcal{P}$ and $\mathcal{Q}$ be the sets of hidden state vectors on which $p_i$ and $q_j$, respectively, are active, and let $\mathcal{F}$ be the set of inputs on which $z_{i,j} > 0$, then it is true that $\mathcal{P} \cap \mathcal{Q} = \mathcal{F}$. The square root in equation 3 prevents activation value explosion when both pre-latents are active. See additional discussion in Appendix C.

The encoder cost per token drops from $\mathcal{O}(Fd)$ to $\mathcal{O}\big(h(m+n)d\big)$ (see Appendix A.2). KronSAE thus reduces FLOPs and parameter count without routing overhead, and is orthogonal to existing sparse decoder kernels (Gao et al., 2025) and thus can be combined with them for end-to-end speed-ups.

# 4 EXPERIMENTS

We train SAEs on the residual streams of Qwen-2.5-1.5B-Base (Yang et al., 2024), Pythia-1.4B-deduped (Biderman et al., 2023), and Gemma-2-2B (Team et al., 2024) language models. Activations are collected on FineWeb-Edu, a filtered subset of educational web pages from the FineWeb corpus (Penedo et al., 2024). We measure reconstruction quality via explained variance (EV),

$$\text{EV} = 1 - \frac{\text{Var}(\mathbf{x} - \hat{\mathbf{x}})}{\text{Var}(\mathbf{x})},$$

so that $1.0$ is optimal, and use automated interpretability pipeline (Bills et al., 2023; Paulo et al., 2024) and SAE Bench (Karvonen et al., 2025) to evaluate properties of SAE features. We aim for the needs of resource-constrained mechanistic interpretability research where efficiency and interpretability are in favor rather than top reconstruction performance, and 100M-820M token budgets are widely adopted (Bussmann et al., 2025; Kharlapenko et al., 2025; Heap et al., 2025; Mudide et al., 2025; Karvonen et al., 2025), so we choose 125M, 500M, 1B and 2B token budgets for the experiments.

Our experiments (see detailed setup in Appendices A and D) address three questions:

1. Does KronSAE maintain EV comparable to baseline SAEs under fixed compute?
2. Which design choices (nonlinearity, $(m, n, h)$) drive EV improvements?
3. How do these choices affect properties and interpretability of learned latents?

## 4.1 ABLATIONS

We employ the iso-FLOPs setup: for each KronSAE variant of dictionary size $F$ we allocate the same amount of FLOPs as was spent for the training of TopK SAE for token budget $T$ and same $F$.

**Reconstruction performance.** As indicated on Figure 1, KronSAE achieves on-par performance with TopK given lower number of trainable parameters and outperforms Matryoshka SAE. The performance gap narrows when increasing the dictionary size, which indicate the potential scalability of our method for large dictionaries. See also result on Gemma-2 2B model in Appendix A.3.

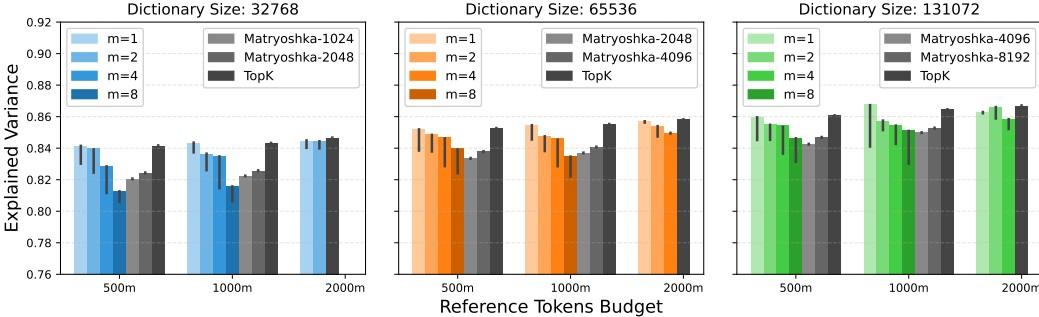

Figure 1: Maximum performance for KronSAE vs. TopK SAE vs. Matryoshka TopK SAE on Qwen-1.5B for different dictionary sizes $F$ and budgets in iso-FLOP setting. KronSAE with lower number of parameters is on-par with the baseline, and the gap narrows with larger dictionary size.

**Decomposition hyperparameters.** We systematically vary the number of heads $h$ and the per-head base dimension $m$ (with $n = F/(mh)$) under the iso-FLOPs setup. From the Figure 2, we conclude that lower $m$ and higher $h$ consistently yields higher reconstruction quality, due to flexibility of pre-latents - as we show in section 5.3, they must be either expressive or fine-grained enough (low $m$, $n$ and high $h$) to efficiently represent the semantics. See also Appendices A.5 and A.8.

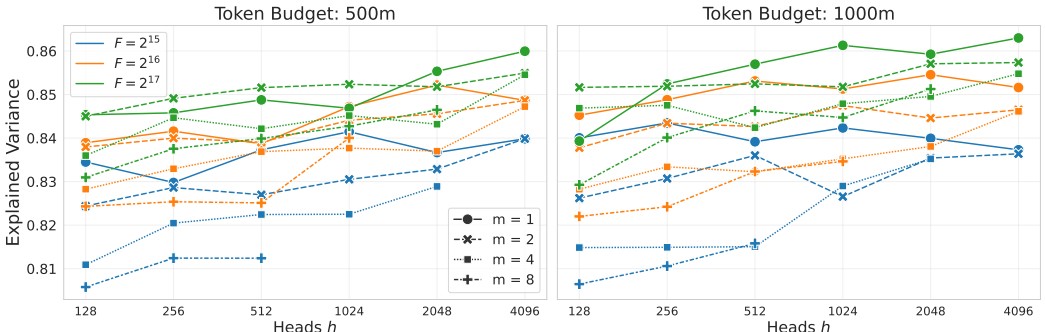

Figure 2: Dependency of EV on head count $h$ (on the x-axis) and base dimension $m$ under 500M and 1B token budgets in iso-FLOPs setup. Higher $h$ and smaller $m$ yield improved reconstruction quality because of higher expressivity of pre-latents to encode semantics and increasing trainable parameters.

**Composition activation.** To isolate the impact of our $\mathrm{mAND}$ operator, we compare it to two simpler interaction kernels: (i) the element-wise product of ReLUs, $\mathrm{ReLU}(u) \cdot \mathrm{ReLU}(v)$, and (ii) the raw product $u \cdot v$. As reported in Table 1, under a 125M training budget, the $\mathrm{mAND}$ variant achieves the highest explained variance. More description of the $\mathrm{mAND}$ is provided in the Appendix C. See also our experiments where we replace TopK with JumpReLU in Appendix A.6.

| Dictionary size | m | n | h | Activation | Explained Variance |
|---|---|---|---|---|---|
| 32768 | 2 | 4 | 4096 | $\mathrm{mAND}(u,v)$ | **0.8336** |
| | | | | $\mathrm{ReLU}(u) \cdot \mathrm{ReLU}(v)$ | 0.8267 |
| | | | | $u \cdot v$ | 0.8237 |
| | 4 | 8 | 1024 | $\mathrm{mAND}(u,v)$ | **0.8220** |
| | | | | $\mathrm{ReLU}(u) \cdot \mathrm{ReLU}(v)$ | 0.8191 |
| | | | | $u \cdot v$ | 0.8143 |
| 65536 | 2 | 4 | 8192 | $\mathrm{mAND}(u,v)$ | **0.8445** |
| | | | | $\mathrm{ReLU}(u) \cdot \mathrm{ReLU}(v)$ | 0.8328 |
| | | | | $u \cdot v$ | 0.8297 |
| | 4 | 8 | 2048 | $\mathrm{mAND}(u,v)$ | **0.8350** |
| | | | | $\mathrm{ReLU}(u) \cdot \mathrm{ReLU}(v)$ | 0.8297 |
| | | | | $u \cdot v$ | 0.8251 |

Table 1: Performance of different composition activations under a budget of 125M tokens.

**Sparsity Analysis.** To evaluate performance across different sparsity budgets ($\ell_0 = \{16, 32, 64, 128\}$), we compare multiple SAE baselines (TopK SAE, Matryoshka SAE, Switch SAE) against their KronSAE variants at dictionary size $F = 2^{16}$ trained on 500 million tokens, under an iso-FLOPs budget matched to the TopK baseline. As shown in Figure 3, KronSAE achieves comparable reconstruction fidelity (measured by explained variance) to all baseline methods across all sparsity levels. Results for smaller dictionary size $F = 2^{15}$ are presented in Appendix A.4.

**Layerwise performance.** Additionally, we evaluate performance across different layers in Qwen-2.5-1.5B. In every case, KronSAE matches the reconstruction quality of the TopK baseline, demonstrating that our Kronecker-factorized encoder maintains its performance regardless of depth. This setup and corresponding results are described in greater detail in Appendix A.6.

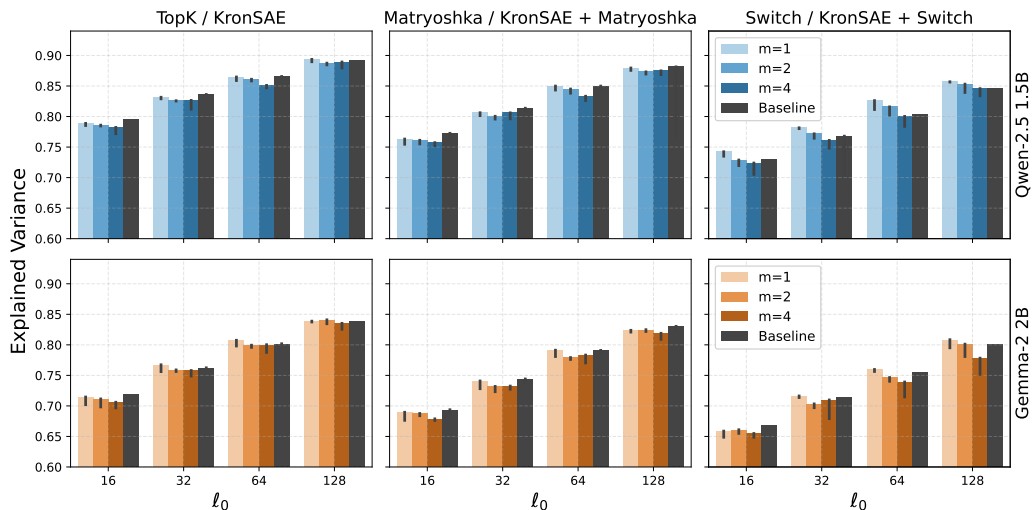

Figure 3: Maximum performance for baselines and their KronSAE modifications for different sparsity levels in iso-FLOP setting. KronSAE variants, despite using fewer trainable parameters, achieve reconstruction quality comparable to or better than the unmodified baselines.

## 4.2 ABSORPTION

The notorious challenge in SAE interpretability is *feature absorption*, where one learned feature becomes a strict subset of another and consequently fails to activate on instances that satisfy the broader concept but not its superset representation (e.g. a "starts with L" feature is entirely subsumed by a "Lion" feature) (Chanin et al., 2024).

Figure 4 reports three absorption metrics measured via SAEBench (Karvonen et al., 2025) across sparsity levels $\ell_0 \in \{16, 32, 64, 128\}$: (1) the *mean absorption fraction*, measuring the proportion of features that are partially absorbed; (2) the *mean full-absorption score*, quantifying complete subsumption events; and (3) the *mean number of feature splits*, indicating how often a single conceptual feature fragments into multiple activations. We use with dictionary size $F = 2^{16}$ and compare TopK SAE, Matryoshka SAE (Bussmann et al., 2025), and their KronSAE version, since they impose different hierarchical priors (see Appendix C for a discussion of how Matryshka SAE and TopK SAE structure differs). Across all $\ell_0$, KronSAE variants consistently reduce first two scores relative to the TopK SAE baseline, while maintaining a similar rate of feature splits.

We attribute KronSAE's improved disentanglement to two complementary design choices:

1. **AND-like behaviour.** By ensuring that post-latent emits only when its more general pre-latent parents are active, we prevent more specific post-latents from entirely subsuming broadly polysemantic one. See additional description of this mechanism in Appendix C.

2. **Head-wise Cartesian decomposition.** Dividing the latent space into $h$ independent subspaces (each with its own $m \times n$ grid of primitive interactions) ensures that specialized concepts (such as "elephant") are confined to a single head and cannot fully absorb more general concepts (such as "starts with E") in another.

Together, these mechanisms produce more monosemantic features, as we verify in the section 5.3, simplifying downstream applications. See results with dictionary size $F = 2^{15}$ in Appendix A.4. We also validate the result on Pythia models and observe the same picture, see the Appendix A.5.

## 5 ANALYSIS

In this section we examine the properties of the latents in KronSAE compared to TopK architecture.

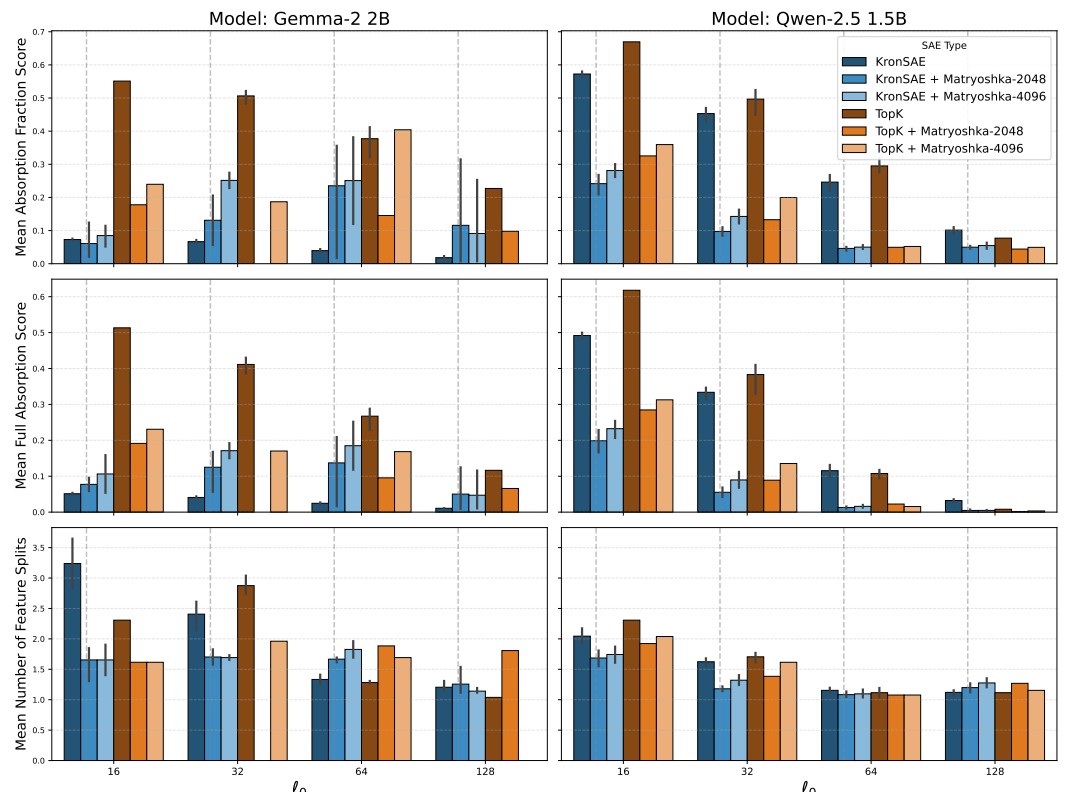

Figure 4: Feature absorption metrics on Qwen-2.5 1.5B and Gemma-2 2B. KronSAE configurations (various $m, n$) exhibit lower mean absorption fractions and full-absorption scores across different $\ell_0$ and selected baselines.

The design of our arhictecture was also inspired by the observation that many features within a single SAE correlate with each other. KronSAE introduces structural bias that forces the post-latents to co-occur with their pre-latents (see also Appendix C). In this section we analyse if it is helpful.

By examining the toy examples with manufactured correlations in data, we show that KronSAE captures these correlations better than TopK. Then we show that KronSAE trained on language indeed moves correlated features within a single head, indicated by higher correlation within head. After that, we show that KronSAE pre-latents interactions are closely resemble the logic AND gate, and its post-latents are notably more interpretable than TopK latents.

## 5.1 TOY MODEL OF CORRELATION

To evaluate how well different sparse autoencoder architectures recover underlying correlation patterns, we construct a controlled experiment using a synthetic, block-structured covariance model. Input vectors $\mathbf{x} \in \mathbb{R}^F$ sampled from a normal distribution (with $\mu = 0, \sigma = 1$). We then perform a Cholesky decomposition $S = LL^\top$ on the covariance matrix $S$ and set $\bar{\mathbf{x}}_{\text{sparse}} = L \, \text{TopK}(\text{ReLU}(\mathbf{x}))$, so that $\bar{\mathbf{x}}_{\text{sparse}}$ exhibits the desired structure.

We train autoencoder (AE) to reconstruct $\bar{\mathbf{x}}_{\text{sparse}}$ following the (Elhage et al., 2022):

$$\hat{\mathbf{x}} = \text{ReLU}(W^\top W \cdot \bar{\mathbf{x}}_{\text{sparse}} + \mathbf{b}). \tag{5}$$

We collect hidden states of dimension $d = 64$ ($W \cdot \bar{\mathbf{x}}_{\text{sparse}}$) from AE and then train TopK SAE and our proposed KronSAE with $F = 256$ and $\text{topk} = 8$ to reconstruct it. After training, we extract the decoder weight matrices $W_{\text{dec}}$ from each SAE, match its latents with the autoencoder latents by solving the quadratic assignment problem (Vogelstein et al., 2015) (see Appendix B for motivation and details), and compute the covariance $C_{\text{dec}} = W_{\text{dec}} W_{\text{dec}}^\top$. Result is shown in Figure 5.

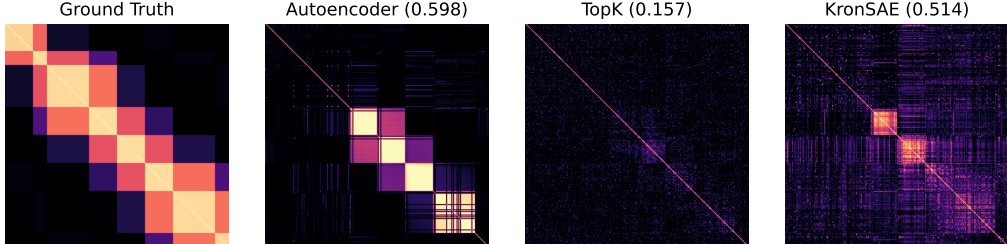

Figure 5: We generate data with covariance matrix that consist of blocks with different sizes on diagonal and off diagonal (left panel). We then examine the decoder-weight covariance $W_{\text{dec}} \cdot W_{\text{dec}}^{\top}$ to assess feature-embedding correlations and compute the RV score to quantify the similarity between learned and ground truth covariance matrices. Second panel show feature embeddings for trained autoencoder $W_{\text{enc}} \cdot W_{\text{enc}}^{T}$. Third panel demonstrates that a TopK SAE recovers these correlation structures weakly, as indicated by relatively low RV coefficient ($0.157$) even after optimal atom matching. In contrast, KronSAE (right panel) more accurately reveals the original block patterns.

To quantify how closely each model's feature correlations mirror the original structure of $S$, we employ the RV coefficient, defined as $RV(S, C) = \text{trace}(SC)/\sqrt{\text{trace}(S^2)\text{trace}(C^2)}$. In our experiments, KronSAE consistently achieves notably higher RV than TopK SAE, indicating that our compositional encoder more faithfully reconstructs the original feature relation. See also additional experiments in section B, where we provide further empirical intuition for how KronSAE identifies correlation structure that more closely align with those present in the data.

## 5.2 Correlations in SAEs Trained on Language

To examine the correlation structure of features learned in our SAE, we have calculated the correlations on 5k texts from the training dataset. For each feature we calculate the mean correlation with features within its head and with all other features, and compare the randomly initialized KronSAE with $m = 4, n = 4$ with the trained one. To isolate the effect of our initialization procedure, we initialize the weights of SAE from the uniform distribution. As shown in Figure 6, correlations are indeed significantly higher within a single head and higher than for random SAE, which suggest that our choice to impose the correlated structure in SAE latents works as intended.

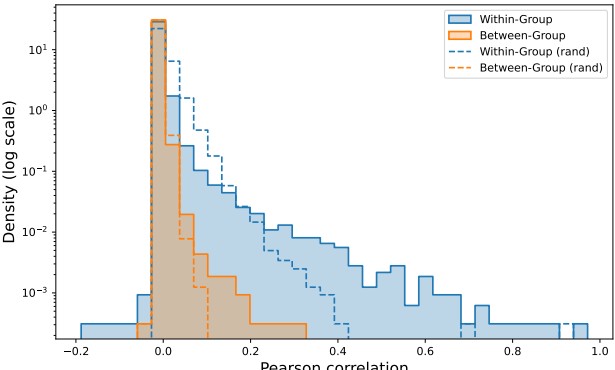

Figure 6: Correlations between features in KronSAE with $m = 4, n = 4$ within a head and with features from other heads. Our design induces higher correlations within a group, which also gets stronger after training, although SAE have also learned correlated features from different heads.

## 5.3 Analysis of Learned Features

In this section we compare KronSAE and TopK SAE in terms of interpretability and feature properties, and we analyze the properties of groups in KronSAE. For this, we choose the 14th layer of Qwen2.5-

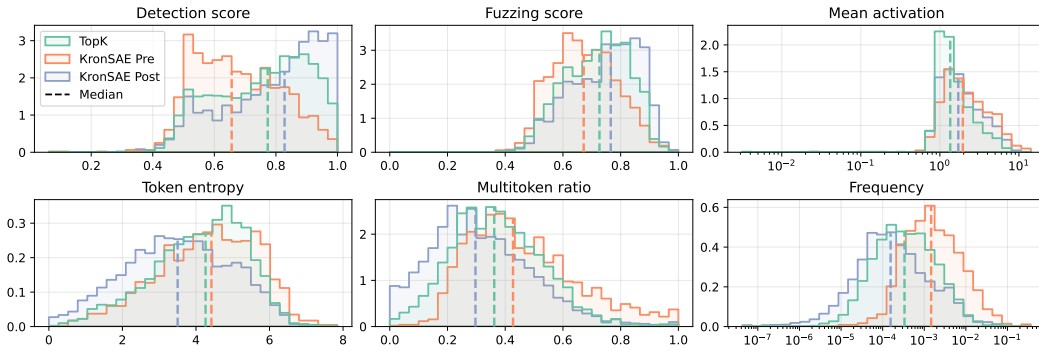

Figure 7: Distribution of properties for TopK SAE and KronSAE ($m = 4, n = 4$) with 32k dictionary size trained on Qwen2.5-1.5B. Pre and Post suffixes denote pre- and post- latents, and y-axis indicate density. Token entropy shows the entropy of the distribution of tokens on which feature has activated, and the multitoken ratio measures how often does feature activate in a single sequence. Our SAE achieves better interpretability scores by learning specialized feature groups, indicated by lower activation frequency and lower variance in activated tokens.

1.5B and a dictionary size of 32k features, of which the first 3072 were selected. KronSAE was chosen with $m = 4, n = 4$. We run for 24M tokens total to collect data. Our interpretation pipeline follows the common methodology: LLM interprets the activation patterns (Bills et al., 2023) and we evaluate obtained interpretations using the *detection* score and the *fuzzing* score Paulo et al. (2024).

For each selected feature, among the standard mean activation value and frequency, we calculate two additional metrics. Low values of *token entropy* suggest that feature activates more frequently on small number of tokens, thus it is token-specific; high value of *multitoken ratio* indicates that feature tends to activate multiple times in a single sentence. We have observed that both these metrics have notable negative correlation with the final interpretability scores and therefore they provide useful signal to assess the potential score without calculating it.

For more details on the data collection and interpretation pipeline, see Appendix D. For additional analysis of properties of learned features , additional comparison with baselines and discussion about tradeoff between reconstruction performance and interpretability, see Appendix E.

**SAE properties and encoding mechanism.** We observe that the features learned by KronSAE are more specific, indicated by lower values of the computed metrics and higher interpretability scores, as shown in Figure 7. Since post-latents are significantly more interpretable than corresponding pre-latents, we hypothesize the hidden mechanism for encoding and retrieval of the required semantics.

By examining activating examples and interpretations of latents, we observe that pre-latents may carry multiple distinct and identifiable modes of activation, such as composition base element 3 in head 23 shown in Table 2, and be very abstract compared to resulting post-latents. Polysemanticity of pre-latents is expected to be a consequence of reduced "working" number of encoder latents, since we decompose the full dictionary size and reduce the encoder capacity.

Thus, we hypothesize that the encoding of specific semantics in our SAE may be done via magnitude, which we validate by examining the activation examples. For the above mentioned pre-latent, the "comparison" part is encoded in the top 75% quantile, while the "spiritual" part is mostly met in the top 25% quantile, and the "geographical" part is mainly encoded in the interquartile range. We also consider but do not investigate the possibility that it may depend on the context, e.g. when the model uses the same linear direction to encode different concepts when different texts are passed to it.

**Semantic retrieval and interpretable interactions.** Heads usually contain a groups of semantically related pre-latents, e.g. in head 136 there are three base elements and one extension covering numbers and ordinality, two extension elements related to geographical and spatial matters, one question-related base and one growth-related extension. Interestingly, most post-latents for this head have higher interpretability score than both its parent pre-latents, which is unusual.

The retrieval happens primarily via the mechanism closely resembling the logical AND circuit, where some pre-latent works as the bearer of multiple semantics, and the corresponding pre-latent (base or extension) works as specifier. An illustrative example is shown in Table 2: we see that the base contains three detectable sub-semantics, and each extension then retrieves the particular semantics.

| Component | Interpretation | Score |
|---|---|---|
| **Base 3** | Suffix "-like" for comparative descriptors, directional terms indicating geographical regions, and concepts related to spiritual or metaphysical dimensions | 0.84 |
| Extension elements and their compositions with base 3 | | |
| **Extension 0** | *Interpretation:* Comparative expressions involving "than" and "as" in contrastive or proportional relationships. | 0.87 |
| | *Composition:* Similarity or analogy through the suffix "-like" across diverse contexts. | 0.89 |
| **Extension 1** | *Interpretation:* Specific terms, names, and abbreviations that are contextually salient and uniquely identifiable. | 0.66 |
| | *Composition:* Medical terminology related to steroids, hormones, and their derivatives. | 0.84 |
| **Extension 2** | *Interpretation:* Spiritual concepts and the conjunction "as" in varied syntactic roles. | 0.80 |
| | *Composition:* Abstract concepts tied to spirituality, consciousness, and metaphysical essence. | 0.93 |
| **Extension 3** | *Interpretation:* Directional and regional descriptors indicating geographical locations or cultural contexts. | 0.84 |
| | *Composition:* Directional terms indicating geographical or regional divisions. | 0.91 |

Table 2: Interactions between composition base element 3 in head 23 and all extension elements in that head. Interaction happens in a way that closely resembles the Boolean AND operation: base pre-latent is polysemous, and the composition post-latent is the intersection, i.e. logical AND between parent pre-latents. See details in Section 5.3.

Other types of interaction may occur, such as appearance of completely new semantics, for example composition between base 3 and extension 1 in Table 2 where medical terminology arises and could not be interpreted as simple intersection between two pre-latents semantics. Another example is a case of head 3 where base 3 has sub-semantics related to technical instruments and extension 2 have semantics related to the posession and necessity, and their combination gives the therapy and treatment semantics which looks more like addition than intersection.

It is a frequent case that post-latent inherit semantics of only one parent, or the impact of another parent is not detectable, which usually happens if parent has a very broad interpretation and low score. However, it requires more sophisticated techniques to properly identify the fine-grained structure of interactions than just looking at the resulting latent descriptions, so we leave it to further work. Despite this, the AND-like gate is a very common behavior. See more examples in Appendix G.

**Geometry of post-latents.** Each post-latent vector has a vector representation in the residual stream represented by the corresponding column in $W_{dec}$, which is the approximation of overcomplete basis vectors we search for when training SAEs. Our architectural design leads to clustering of feature embeddings so that post-latents produced by same head, base or a extension elements are grouped in a tight cluster, and the geometry is dependent on hyperparameters $h, m, n$ we choose, which is expected and may be useful for further applications such as steering. See more details in Appendix E.

## 6 CONCLUSION AND FUTURE WORK

We introduce **KronSAE**, a sparse autoencoder architecture design that combines head-wise Kronecker factorization of latent space with a approximation of logical AND via mAND nonlinearity. Our approach allows to efficiently train interpretable and compositional SAE, especially in settings with limited compute budget or training data, while maintaining reconstruction fidelity and yielding more interpretable features by utilizing their correlations. Our analysis links these gains to the complementary effects of compositional latent structure and logical AND-style interactions, offering a new lens on how sparsity and factorization can synergise in representation learning.

**Limitations.** KronSAE introduces tradeoff between interpretability, efficiency and reconstruction performance, and due to reduced number of trainable parameters it is expected to lag behind TopK SAE at large budgets. Our evaluation is limited to mid-sized transformer models and moderate dictionary sizes; however, the main bottleneck there might be not the SAE itself, but the infrastructure required to handle these setups and the model inference.

**Future Work.** We identify three directions for extending this work: (i) *Transcoding.* Treat transcoders (Dunefsky et al., 2024) as implicit routers of information and investigate alternative logical gating functions (e.g. XOR or composite gates) to improve interpretability and circuit analysis. (ii) *Crosscoding.* Generalize KronSAE to a crosscoder setting (Lindsey et al., 2024) uncover interpretable, cross-level compositionality via logic operations. (iii) *Dynamic Composition.* Explore learnable tuning of both the number of attention heads and their dimensionality, enabling fine-grained decomposition into groups of correlated features at varying scales.

## ETHICS STATEMENT

While interpretability research has dual-use potential, our method operates within the ethical boundaries of the underlying models and aims to advance responsible AI development through better model understanding. We analyze activations from publicly available language models (Qwen-2.5-1.5B, Pythia-1.4B, and Gemma-2-2B) gathered on FineWeb-Edu datasets, which excludes the unreported harmful content. We declare no conflicts of interest and maintain transparency about limitations, including potential artifacts from LLM-based interpretation as noted in Appendices D.3 and I.

## REPRODUCIBILITY STATEMENT

We have taken several measures to ensure the reproducibility of our results. We use publicly available models (Qwen, Gemma, Pythia families) and training dataset (FineWeb-Edu) in our experiments. Section 4 and Appendix A provide detailed description of SAE training procedure and hyperparameter configuration. Our complete implementation is available in the supplementary materials, containing the training code, interpretation pipeline and analysis of the results. Appendix H includes simplified implementation of KronSAE that might be easily integrated into existing training codebases, while Appendix D details the interpretability analysis methodology with precise evaluation protocols.

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

# A  ADDITIONAL DETAILS AND RESULTS

## A.1  EXPERIMENTAL SETUP

**Training details.**   All SAEs are optimized using AdamW with an initial learning rate of $8 \times 10^{-4}$, a cosine learning-rate schedule with a minimum LR of $1 \times 10^{-6}$, and a linear warmup for the first 10% of total training steps, auxiliary loss penalty equal to $0.03125$. We use a global batch size of 8,192. We sweep over dictionary (latent) sizes of $F = 2^{15}$, $F = 2^{16}$, and $F = 2^{17}$ features. For our KronSAE variant, we further sweep the number of heads $h$ and the per-head dimensions $m$ and $n$ such that $h \cdot m \cdot n$ equals the desired dictionary size. Regularization weights and auxiliary loss coefficients are kept constant throughout the runs to isolate the impact of architectural choices.

For all experiments, we spent about 330 GPU days on NVIDIA H100 80GB GPUs, including preliminary research.

**SAE.**   For all experiments on Qwen-2.5, we train each SAE on activations from layer 14. Also for Pythia-1.4B we use layer 14 and for Gemma-2-2B we take activations from layer 12. For most of our experiments, we use sparsity level of $\ell_0 = 50$ non-zero activations per token.

**Initialization.**   As observed by Gao et al. (2025), initializing the decoder as the transpose of the encoder ($W_{\text{dec}} = W_{\text{enc}}^{\top}$) provides a strong metric improvement. We adopt this strategy within KronSAE by partitioning $W_{\text{enc}}$ into $h$ head-wise blocks of shapes $m \times d$ and $n \times d$, denoted $\{P_i, Q_i\}_{i=1}^{h}$. For each head $k$, we define its decoded rows via a simple additive composition:

$$C_k[i, j] = P_{k,i} + Q_{k,j}, \quad i = 1, \ldots, m, \ j = 1, \ldots, n.$$

Finally, flattening the matrices $\{C_k\}$ yields full decoder weight matrix $W_{\text{dec}} \in \mathbb{R}^{F \times d}$.

**Matryoshka and Kron-based version.**   For Matryoshka SAE of dictionary size $F$ we adopt the following experimental setup. We use most training settings from Training details. For Matryoshka SAE Bussmann et al. (2025) we define the parameter of dictionary group $S = [2^k, 2^k, \cdots 2^{k+i} \cdots 2^n]$, where $k < n$ and $\sum_{s \in S} s = F$. This is equivalent to nested sub-SAEs with dictionary sizes $\mathcal{M} = \{2^k, 2^k + 2^k, 2^k + 2^k + 2^{k+1}, \cdots, F\}$ and in our work we define this SAE as **Matryoshka-$2^k$**. Training loss for Matryoshka is defined as follows:

$$\mathcal{L}(\mathbf{f}) = \sum_{m \in \mathcal{M}} ||\mathbf{x} - \mathbf{f}_{0:m} W_{0:m}^{dec} + b_{dec}||_2^2 + \alpha \mathcal{L}_{aux}, \tag{6}$$

where $\mathbf{f}$ is latent vector and $\mathcal{L}_{aux}$ is auxiliary loss used in Gao et al. (2025).

For Kron-based Matryoshka SAE we use same setup and same initialization from Initialization section. We choose $m$ and $n$ so that $F \mod mn = 0$, ensuring that no heads are shared between groups.

**Switch SAE and Kron-based version.**   For the Switch SAE architecture proposed by Mudide et al. (2025), we configure all experiments with 8 experts and use identical initialization from KronSAE. In the Kron-based variant of this SAE, we distribute the heads equally across all experts, resulting in $F/(8mn)$ heads per expert. Hence we have less parameters in every expert and therefore we also reduce FLOPs for encoder.

## A.2  FLOPS CALCULATION AND EFFICIENCY

For TopK SAE and KronSAE we compute FLOPs in the following way:

$$\begin{aligned} \text{FLOPS}_{\text{TopK}}(d, F, k) &= dF + kd, \\ \text{FLOPS}_{\text{KronSAE}}(d, m, n, h, k) &= dh(m + n) + mnh + kd \approx dh(m + n) + kd. \end{aligned} \tag{7}$$

We calculate FLOPs for most effective variant of TopK where we perform vector matrix multipication only for nonzero activations, while encoder still requires dense matrix multiplication.

We have also measured the wallclock time for forward and backward to examine the scaling. Figure 8 reports scaling for different hidden dimension sizes.

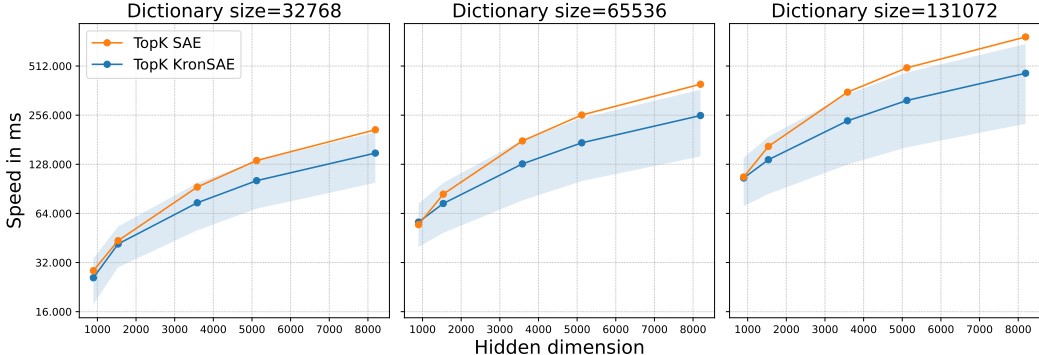

Figure 8: Speed comparision of TopK SAE with KronSAE across different hidden dimensionss. We can see that KronSAE have better scaling properties than SAE with default encoder architecture.

## A.3 SCALING ON GEMMA-2 2B

To examine the method's generality, we conducted additional reconstruction experiments on Gemma-2 2B under iso-FLOPs settings. As shown in Figure 9, our method achieves comparable or improved reconstruction performance across the evaluated compute budgets.

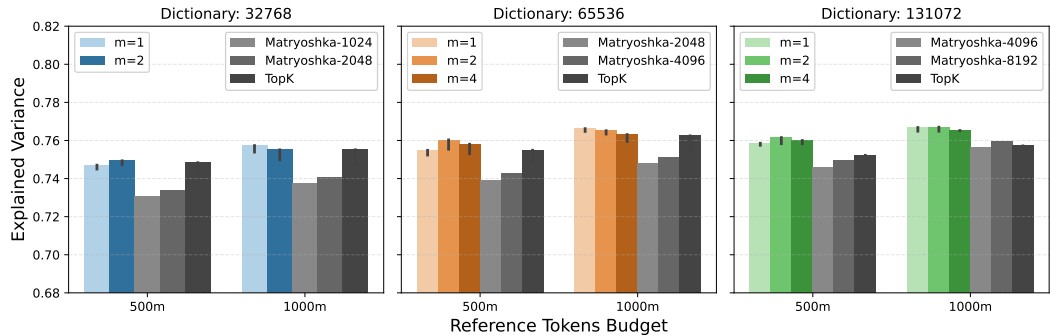

Figure 9: Performance comparision of KronSAE and TopK SAE under a fixed iso-FLOPs budget. Across sparsity settings, KronSAE typically matches TopK's reconstruction performance and in some cases slightly outperforms it, while using much fewer trainable parameters.

## A.4 SMALLER DICTIONARY SIZE

To complement our larger-scale experiments, we further evaluate KronSAE's performance on a smaller dictionary size $F = 2^{15}$ with varying $\ell_0 = \{16, 32, 64, 128\}$ and **equal token budget**.

**Sparsity.** Following the experimental setup described in Section 4.1, we compare KronSAE against established baselines including TopK SAE, Matryoshka SAE, and Switch SAE . As shown in Figure 10, KronSAE consistently matches or exceeds the reconstruction performance of baseline architectures across all tested sparsity levels, while achieving these results with substantially fewer trainable parameters and FLOPs.

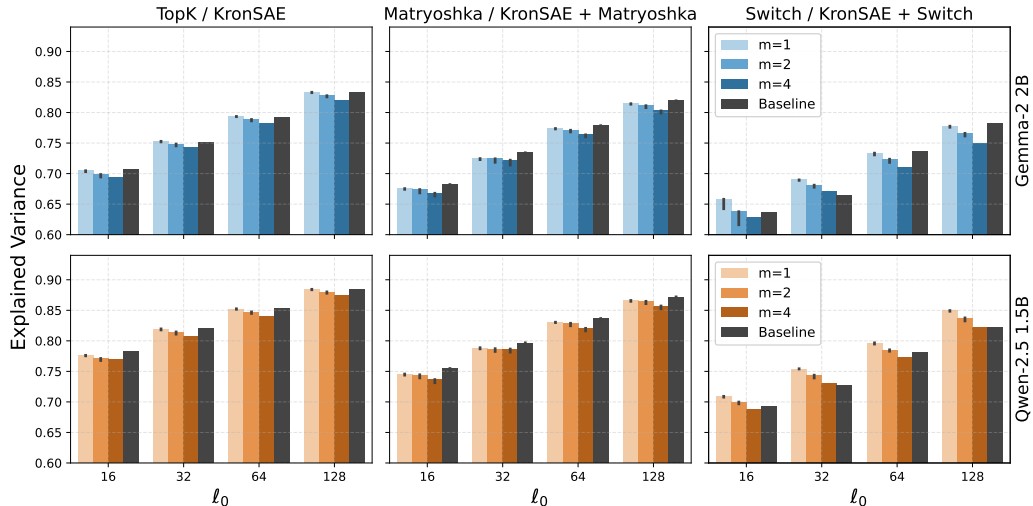

Figure 10: Performance comparision of KronSAE and TopK SAE under a fixed token budget. Across sparsity settings, KronSAE typically matches TopK's reconstruction performance and in some cases slightly outperforms it, while using much fewer trainable parameters and smaller FLOPs budget.

**Feature absorption.** We extend our feature absorption analysis from Section 4.2 to the smaller dictionary size, employing two distinct grouping schedules: $S_1 = [2048, 2048, 4096, 8192, 16384]$ and $S_2 = [1024, 1024, 2048, 4096, 8192, 16384]$. As shown in Figure 11 KronSAE modification demonstrates better performance, reducing feature absorption metrics.

## A.5 PYTHIA SUITE

For Pythia-1.4B we train all SAEs on the 12th transformer layer with a budget of 125M tokens. As reported in Table 3, KronSAE achieves performance comparable to TopK SAE with increased number of heads.

| Dictionary | SAE | Mean | Max |
|---|---|---|---|
| 32k | TopK | | 0.793 |
| | KronSAE | 0.783 | 0.793 |
| 65k | TopK | | 0.802 |
| | KronSAE | 0.795 | 0.805 |
| 131k | TopK | | 0.801 |
| | KronSAE | 0.800 | 0.810 |

Table 3: Performance of Pythia-1.4B at 125M budget. At larger dictionary size and fixed training budget KronSAE outperforms TopK SAE.

We conducted additional experiments with smaller Pythia models and trained KronSAE at the middle layers with 512 heads. Table 4 reports results for 125M budget on 65k and 262k dictionary sizes.

In section 4.2 we have analysed whether KronSAE achieves lower absorption score and have answered affirmatively. We also compare the results for Pythia 1.4B model and validate the improvements, as reported in the Table 5.

These results confirm that our compositional architectures improves the absorption score and feature consistency across various models from different families.

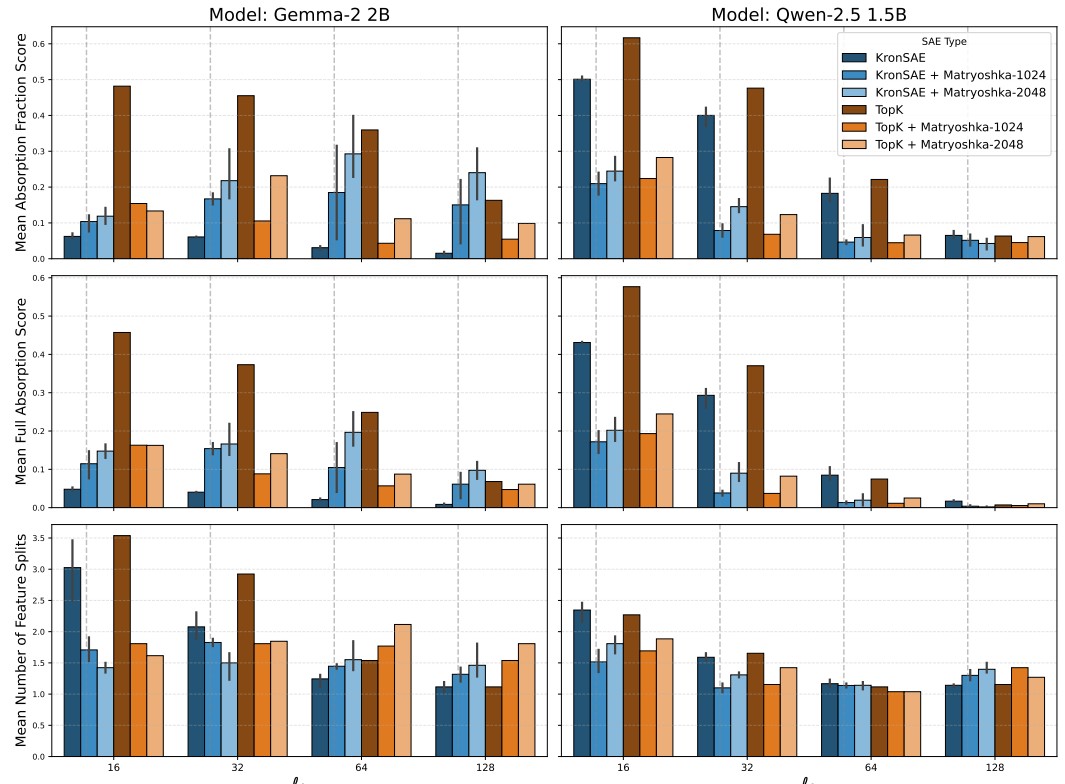

Figure 11: Feature absorption scores comparison of KronSAE vs baseline methods (TopK SAE and Matryoshka SAE) and a Kron-enhanced Matryoshka SAE variant.

| Model | 70M (d=512) | | 160M (d=768) | | 410M (d=1024) | |
|---|---|---|---|---|---|---|
| Dictionary | 65k | 256k | 65k | 256k | 65k | 256k |
| $m = 1$ | 0.893 | 0.892 | 0.856 | 0.855 | 0.834 | 0.835 |
| $m = 2$ | 0.896 | 0.897 | 0.859 | 0.859 | 0.832 | 0.841 |
| $m = 4$ | 0.894 | 0.899 | 0.857 | 0.859 | 0.828 | 0.839 |
| $m = 8$ | 0.896 | 0.899 | 0.856 | 0.862 | 0.827 | 0.846 |
| TopK | 0.905 | 0.903 | 0.870 | 0.867 | 0.843 | 0.847 |

Table 4: Performance of SAEs on 70M, 160M and 410M Pythias with varying hidden dimensionality.

## A.6 COMPARISON WITH JUMPRELU

We provide experiments to compare KronSAE with an alternative activation mechanism, JumpReLU, and report explained variance under three sparsity levels in Table 6.

Replacement of TopK with JumpReLU within KronSAE leads to a degraded performance relative to both JumpReLU SAE and KronSAE with TopK, also with degraded scaling over $\ell_0$. This suggests that the architectural advantages of KronSAE interact most effectively with TopK's behaviour. Whether an alternative activation function can improve on this remains a topic for future work.

| Model | SAE | $\ell_0 = 16$ | $\ell_0 = 32$ | $\ell_0 = 64$ | $\ell_0 = 128$ | $\ell_0 = 256$ |
|---|---|---|---|---|---|---|
| Pythia | TopK | 0.445 | 0.233 | 0.058 | 0.006 | 0.004 |
| | KronSAE | 0.244 | 0.129 | 0.033 | 0.007 | 0.003 |

Table 5: Absorption score calculated for Pythia 1.4B model. KronSAE shows lower score due to structured latent space and hierarchy between pre-latents and post-latents.

| Model Variant | $\ell_0 = 32$ | $\ell_0 = 50$ | $\ell_0 = 64$ |
|---|---|---|---|
| TopK | 0.809 | 0.837 | 0.852 |
| JumpReLU | 0.813 | 0.838 | 0.844 |
| KronSAE (TopK) | 0.814 | 0.840 | 0.853 |
| KronSAE (JumpReLU) | 0.790 | 0.817 | 0.828 |

Table 6: Performance of SAEs with JumpReLU and TopK activations. Since we have floating sparsity controlled via l0 penalty coefficient, we performed a sweep over various sparsity levels, fitted a parabola to the resulting data as a function of $\ell_0$, and evaluated it on those sparsity levels. In contrast, TopK and KronSAE (with TopK) were trained using fixed, predefined sparsity levels.

### A.7 PERFORMANCE ACROSS LAYERS

For this experiment we fix the dictionary size to $F = 2^{15}$ and use the same hyperparameters as in the main experiments. As the Figure 12 shows, KronSAE is on-par with TopK at every depth, demonstrating that structured encoder's reconstruction quality is robust to layer choice.

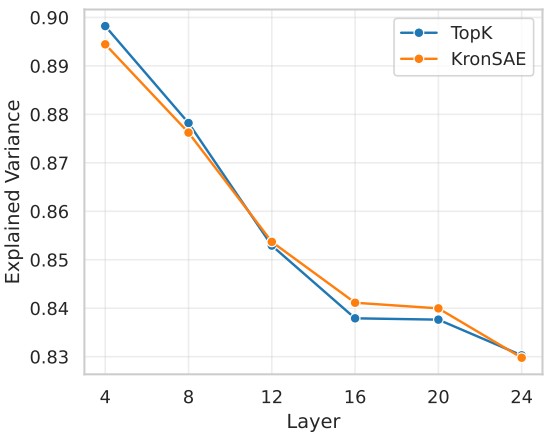

Figure 12: EV across layers of Qwen-2.5-1.5B, demonstrating that KronSAE matches TopK performance regardless of depth.

### A.8 CHOICE OF $m, n, h$

We derive the following guidelines to train KronSAE: one should minimize the $m$ and maximize the $h$ to improve the reconstruction performance, and search for the most expressive configuration from feasible ones. Since $m$ has more impact on EV, one should start from $m = 2$ in the search process, since it gives improved computational performance with on-par EV with full TopK training.

## B MORE RESULTS ON SYNTHETIC

In this section we present the motivation behind our matching algorithm and additional results.

## B.1 FEATURE MATCHING AS QUADRATIC PROBLEM

Suppose that we have two sets of feature embeddings from different models represented as matrices $X, Y \in \mathbb{R}^{F \times d}$, where $F$ is the number of features and $d$ is the dimensionality of feature embeddings. Our task is to find the optimal assignment between features from $Y$ to features from $X$ so that this assignment would satisfy some considerations.

The standard approach is to solve linear assignment problem - find a permutation matrix $\Pi$ subject to minimizing the trace$(C^T\Pi)$, where $C$ is a cost matrix defined as pairwise distance between features $C_{i,j} = d(X_i, Y_j)$. Standard algorithm for solving it in context of sparse dictionary learning is a Hungarian algorithm (Paulo & Belrose, 2025; Balagansky et al., 2025; Fel et al., 2025).

This linear problem is that it only considers pairwise information between features while ignoring the global dependencies between features within $X$ and $Y$ sets separately, e.g. clusters should map to clusters, and the linear problem does not internalize this information. Our observation of correlations between features naturally requires to search for assignment that would take this information into account: we seek for a permutation $\Pi$ that would give best *global* alignment, measured as the Frobenius norm of $X^T\Pi Y$. So the objective becomes:

$$\max_{\Pi} \|X^T\Pi Y\|_F^2 = \max_{\Pi} \text{trace}(\Pi^T X X^T \Pi Y Y^T), \qquad (8)$$

where global feature structure is explicitly encoded in the matrices $XX^T$ and $YY^T$.

An efficient algorithm to solve the quadratic assignment problem is Fast Approximate Quadratic Programming (FAQ) method (Vogelstein et al., 2015) that initially was designed for graphs matching: given the adjacency matrices $A$ and $B$, it minimizes the trace$(A\Pi B^T \Pi^T)$, where $\Pi$ are relaxed from permutation matrices to the set of doubly stochastic matrices (Birkhoff polytope). In our case we define $A = XX^T$ and $B = YY^T$, and since we do not want to minimize the cost but rather maximize the similarity, we solve for the reversed objective:

$$\max_{\Pi}(XX^T\Pi(YY)^T\Pi^T) = \max_{\Pi} \text{trace}(\Pi^T X X^T \Pi Y Y^T), \qquad (9)$$

which is the same as equation 8, and this formulation preserves global dependencies because the contribution of assigning $Y_j$ to $X_i$ depends on all other assignments through the cross-terms in the quadratic form. Listing 1 shows the implementation of this matching procedure.

Listing 1: Implementation of FAQ algorithm for quadratic feature assignment problem.

```python
def feature_matching(A, B, max_iter):
    F, d = A.shape[0]
    G_A, G_B = A @ A.T, B @ B.T
    P = np.ones((F, F)) / F # Barycenter initialization

    # Frank-Wolfe iterations
    for _ in range(max_iter):
        grad = 2 * G_A @ P @ G_B  # Gradient for maximization
        r, c = linear_sum_assignment(-grad)  # Solve LAP
        Q = np.zeros_like(P)
        Q[r, c] = 1

        # Compute optimal step size
        D = Q - P
        b, a = np.trace(grad.T @ D), np.trace(G_A @ D @ G_B @ D.T)

        if abs(a) < 1e-12:
            alpha = 1.0 if b > 0 else 0.0
        elif a < 0:  # Concave case
            alpha = np.clip(-b/(2*a), 0, 1)
        else:         # Convex case
            alpha = 1.0 if b > 0 else 0.0

        P_new = P + alpha * D
        P = P_new
```

```
27      # Project to permutation matrix
28      r, c = linear_sum_assignment(-P)
29      P_final = np.zeros((F, F))
30      P_final[r, c] = 1
31
32      return P_final, P_final @ B
```

## B.2  ADDITIONAL RESULTS

As shown in Figure 13, KronSAE maintains correlation structure that is heavier than in TopK and is better aligned with ground truth covariance matrix.

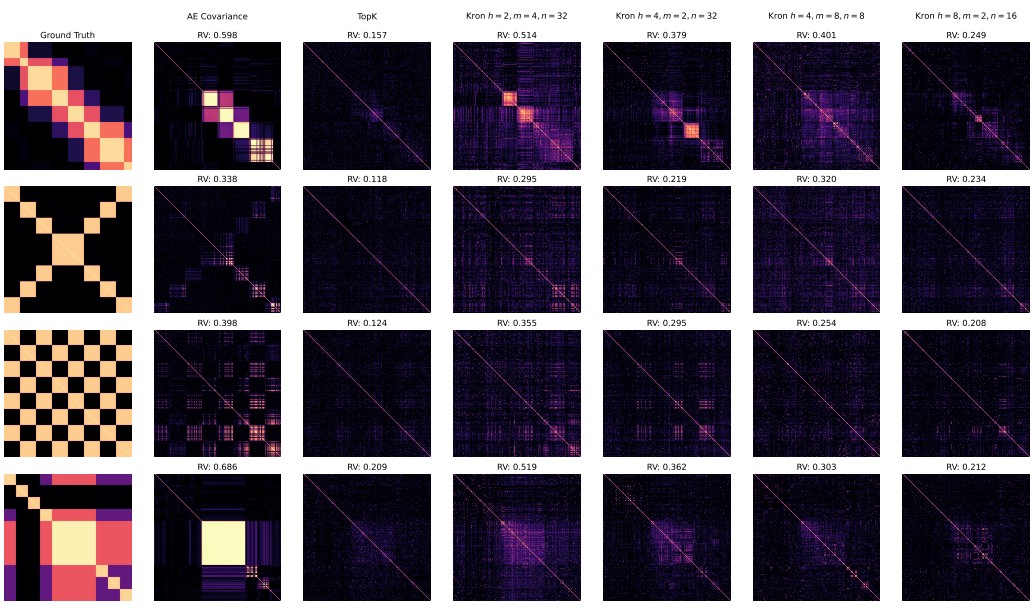

Figure 13: Examples of patterns learned in autoencoder TopK and KronSAE variants after we apply the improved matching scheme. KronSAE have learned patterns that more closely resemble the underlying ground truth structure, and with increasing number of heads (more fine-grained structure) it approaches the TopK SAE.

Table 7 indicate that structure learned by KronSAE is more diverse as indicated by standard deviation of the corresponding values, and is also more aligned with the ground truth covariance according to higher RV scores and lower differences ($\Delta$) between properties of ground truth and learned matrix.

| Model | RV coeff. | Effective rank | Mean corr. | Rank $\Delta$ |
|---|---|---|---|---|
| AE | $0.59 \pm 0.17$ | $63.3 \pm 0.3$ | $0.13 \pm 0.04$ | $47.9$ |
| TopK | $0.12 \pm 0.02$ | $58.5 \pm 1.2$ | $0.08 \pm 0.01$ | $43.2$ |
| Kron $h=2, m=4, n=32$ | $0.22 \pm 0.07$ | $53.8 \pm 3.4$ | $0.08 \pm 0.01$ | $38.5$ |
| Kron $h=4, m=2, n=32$ | $0.3 \pm 0.12$ | $46.0 \pm 7.8$ | $0.10 \pm 0.03$ | $30.7$ |
| Kron $h=4, m=8, n=8$ | $0.23 \pm 0.08$ | $48.8 \pm 5.0$ | $0.09 \pm 0.02$ | $33.5$ |
| Kron $h=8, m=2, n=16$ | $0.17 \pm 0.04$ | $55.9 \pm 1.7$ | $0.07 \pm 0.01$ | $40.6$ |

Table 7: Results with improved matching scheme, computed for 8 different covariance setups. Higher RV coefficients between ground truth and learned matrices and lower $\Delta$ between properties of these matrices indicate that KronSAE is more variable across different setups and is better aligned with ground truth correlation structures.

Together, these results additionally validate improved covariance reconstruction in KronSAE.

## C    mAND AS A LOGICAL OPERATOR AND KRONSAE AS LOGICAL SAE

**AND-like mechanism and hierarchy.** Suppose we have $\mathbf{u}^k = P^k\mathbf{x}$ and $\mathbf{v}^k = Q^k\mathbf{x}$. KronSAE can be described in two equivalent ways:

1. Applying the ReLU to latents $\mathbf{u}$, $\mathbf{v}$, then applying the Kronecker product and square root.

2. Applying the mAND kernel that creates the matrix as in equation 3 and flattening it.

These approaches are complementary. To formally understand how they induce the AND-like mechanism and hierarchy, consider base and extension pre-latent vectors $\mathbf{p} = \text{ReLU}(\mathbf{u})$ and $\mathbf{q} = \text{ReLU}(\mathbf{v})$. Each $p_i$ is a $i$th base pre-latent activation, and $q_j$ is $j$th extension pre-latent activation. Kronecker product $\mathbf{p} \otimes \mathbf{q}$ creates the vector $(p_1 * q_1, \ldots, p_1 * q_m, p_2 * q_1, \ldots, p_n * q_m)$ of the activations of post-latents. Then there is two situations:

1. Post-latent is active $\implies$ both pre-latents activations are positive.

2. Post-latent is inactive $\implies$ at least one of pre-latent activations is zero.

Hence post-latent is active only when both pre-latents are active. Fix some post-latent and its corresponding pre-latents, and suppose that $\mathcal{P}, \mathcal{Q}, \mathcal{F}$ are the sets of input vectors from the hidden state space (passed to SAE) on which base pre-latent, extension pre-latent and post-latent are active. Then it is true that $\mathcal{F} = \mathcal{P} \cap \mathcal{Q}$, meaning that pre-latents must be broader and polysemous to encode multiple semantics of emitted post-latents. We validate this behaviour qualitatively in section 5.3, although in the same section we describe that apparently other types of interactions also presented.

**Matryoshka and Kron hierarchy.** In contrast to TopK SAE, Matryoshka loss imposes different kind of hierarchy by dividing the dictionary into groups of $G_1, \ldots, G_k$ latents where each is of different level of granularity. Namely, first $G_1$ latents are the most broad and abstract, next $G_2$ latents add more fine-grained semantics, lowering the level of abstraction, and so on. This type of structure imposed by specific loss function - increasing the level of granularity must decrease the reconstruction error, and the lowest level of $G_1$ features must also maintain good reconstruction quality - and does not strictly demand some kind of conditional activation of features, while KronSAE imposes two-level AND-like hierarchy via architectural design (Kronecker product). As we have two different mechanisms of feature hierarchy (from encoding mechanism and loss function design) we can combine it, as shown in Section 4.1, 4.2 and Appendix A.4, to combine properties of both approaches.

**Visual intuition.** We also compare our mAND to existing $\text{AND}_{\text{AIL}}$ (Lowe et al., 2021). Since our objective is to drive each atom toward a distinct, monosemantic feature, we found that tightening the logical conjunction encourages sharper feature separation. Moreover, by using the geometric mean ($\sqrt{pq}$) rather than a simple product or minimum, mAND preserves activation magnitudes and prevents post-latent activation to be exploded when both $p, q$ are positive. A visual comparison of mAND and $\text{AND}_{\text{AIL}}$ appears in Figure 14.

## D    FEATURE ANALYSIS METHODOLOGY

We analyze learned features using an established pipeline Bills et al. (2023); Paulo et al. (2024) consisting of three stages: (1) statistical property collection, (2) automatic activation pattern interpretation, and (3) interpretation evaluation. The following subsections detail our implementation.

### D.1    DATA COLLECTION

Our collection process uses a fixed-size buffer $B = 384$ per feature, continuing until processing a predetermined maximum token count $T_{max}$. The procedure operates as follows:

Initial processing batches generate large activation packs of 1M examples, where each example comprises 256-token text segments. When encountering feature activations, we add them to the buffer, applying random downsampling to maintain size $B$ when exceeding capacity. This approach enables processing arbitrary token volumes while handling rare features that may require extensive sampling.

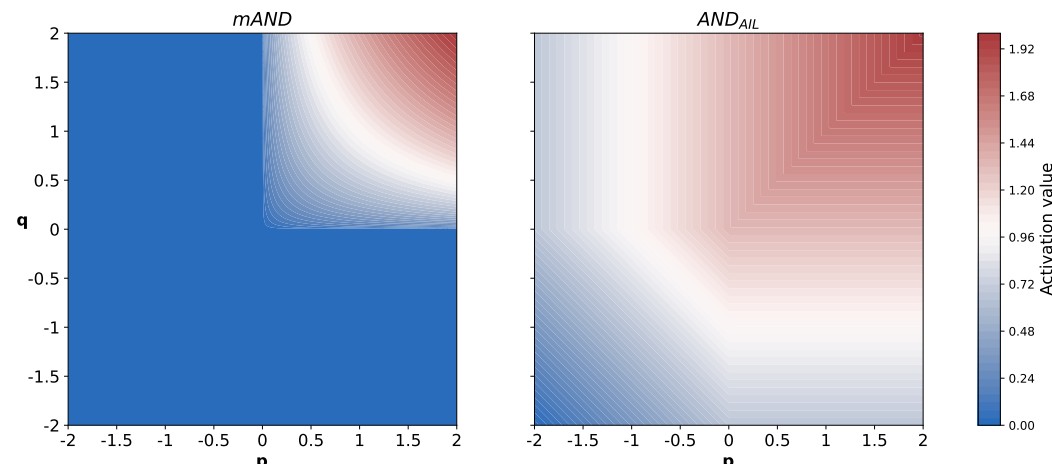

Figure 14: Comparison of the smooth $\mathrm{mAND}$ operator against the $\mathrm{AND_{AIL}}$ (Lowe et al., 2021).

During collection, we compute online statistics including activation minimums, maximums, means, and frequencies. Post-processing yields two key metrics: token entropy and multitoken ratio. The token entropy is calculated as:

$$\text{token entropy} = -\sum_{i=0}^{n} p_i \cdot \log(p_i), \quad p_i = \frac{\text{activations of token i}}{\text{total amount of activations}}, \tag{10}$$

where $n$ represents unique activated tokens. The multitoken ratio is:

$$\text{multitoken ratio} = \frac{1}{b} \sum_{i=0}^{b} \frac{\text{number of activations in sequence } i}{\text{total tokens in sequence } i}, \tag{11}$$

with $b < B$ denoting collected context examples per feature.

We then segment examples using a 31-token context window (15 tokens before/after each activation), potentially creating overlapping but non-duplicated examples. Features with high multitoken ratio may have number of examples significantly exceeding $B$.

A separate negative examples buffer captures non-activating contexts. Future enhancements could employ predictive modeling (e.g., using frequent active tokens) to strategically populate this buffer with expected-but-inactive contexts, potentially improving interpretation quality.

### D.2 FEATURE INTERPRETATIONS

For each feature, we generate interpretations by sampling 16 random activation examples above the median activation quantile and presenting them to Qwen3 14B (Yang et al., 2025) (AWQ-quantized with reasoning enabled). The model produces concise descriptions of the activation patterns. Empirical observations suggest reasoning mode improves interpretation quality, though we lack quantitative measurements. This aligns with findings in (Paulo et al., 2024), which compared standard one-sentence responses with Chain-of-Thought outputs, making model reasoning an interesting direction for future research.

The interpretation process uses the system prompt presented in a Figure 15. User prompts include all special characters verbatim, as some features activate specifically on these characters. A representative (slightly abbreviated) user prompt example is presented on Figure 16.

```
You are a meticulous AI researcher conducting an important
investigation into patterns found in language.  Your task is
to analyze text and provide an explanation that thoroughly
encapsulates possible patterns found in it.

Guidelines:

You will be given a list of text examples on which special
words are selected and between delimiters like «this».  If a
sequence of consecutive tokens all are important, the entire
sequence of tokens will be contained between delimiters «just
like this».  How important each token is for the behavior is
listed after each example in parentheses.

- Your explanation should be a concise STANDALONE PHRASE that
describes observed patterns.
- Focus on the essence of what patterns, concepts and
contexts are present in the examples.
- Do NOT mention the texts, examples, activations or the
feature itself in your explanation.
- Do NOT write "these texts", "feature detects", "the
patterns suggest", "activates" or something like that.
- Do not write what the feature does, e.g.  instead of
"detects heart diseases in medical reports" write "heart
diseases in medical reports".
- Write explanation in the last line exactly after the
[EXPLANATION]:
```

Figure 15: System prompt for feature interpretations.

```
Examples of activations:

Text:  ’  Leno«,» a San Francisco Democrat«, said in a
statement.»’
Activations:  ’  said (22.74), statement (27.84), in (27.54)’

Text:  ’  city spokesman Tyler Gamble« said in an» email.’
Activations:  ’  said (2.92), in (12.81), an (14.91)’

Text:  ’  towpath at Brentford Lock.  «Speaking» on BBC
London 94’
Activations:  ’Speaking (3.48)’

Text:  ’  Michelle, a quadriplegic,« told» DrBicuspid.com’
Activations:  ’  told (4.05)’

Text:  ’  CEO Yves Carcelle« said in a statement».’
Activations:  ’  said (19.64), in (29.09), statement (29.39)’
```

Figure 16: Example of user prompt passed to LLM. This feature with 16 examples received the interpretation "*Structural elements in discourse, including speech attribution, prepositional phrases, and formal contextual markers*" with a detection score of 0.84 and fuzzing score of 0.76.

### D.3 EVALUATION PIPELINE

We evaluate interpretations using balanced sets of up to 64 positive (activation quantile > 0.5) and 64 negative examples, employing the same model without reasoning to reduce computational costs. When insufficient examples exist, we maintain class balance by equalizing positive and negative counts. The evaluation uses modified system prompts from (Paulo et al., 2024), with added emphasis on returning Python lists matching the input example count exactly. We discard entire batches if responses are unparseable or contain fewer labels than the number of provided examples.

We calculate two scores.

**Detection Score:** After shuffling positive/negative examples, we present up to 8 unformatted text examples per batch to the model. The model predicts activations (1/0) for each example, generating up to 128 true/predicted label pairs. The score calculates as:

$$\text{score} = \frac{1}{2} \left( \frac{\text{correctly predicted positives}}{\text{total positives}} + \frac{\text{correctly predicted negatives}}{\text{total negatives}} \right). \tag{12}$$

**Fuzzing Score:** We «highlight» activated tokens on sampled examples, from which 50% are correctly labeled positive examples, 25% are mislabeled positive examples, and 25% are randomly labeled negative examples. We present batches of up to 8 examples and the model identifies correct/incorrect labeling, with scoring following Equation 12.

## E ADDITIONAL FEATURE ANALYSIS RESULTS

**Feature property correlations.** Our analysis reveals significant correlations between feature properties and interpretability scores (Figure 17). Notably, token entropy and mean activation show substantial correlations with interpretability scores, suggesting their potential as proxies for assessing feature quality without running the full interpretation pipeline. These findings are based on analysis of the first 3072 features from 32k TopK and KronSAE (m=4, n=4) trained on 24M tokens, warranting further validation with larger-scale studies.

**Pre-latent to post-latent relationships.** We investigate how post-latent properties correlate with various combinations of pre-latent properties, including individual values, means, products, and the $\mathrm{mAND}$ operation (product followed by square root). Figure 18 demonstrates that post-latent multitoken ratio, token entropy, and frequency show stronger correlations with pre-latent products or $\mathrm{mAND}$ values than with individual pre-latent properties or their means.

**Basis geometry.** As noted in Section 5.3, latent embeddings primarily exhibit clustering within their originating groups (head, base, extension). With the support of observations reported in Sections 4.1 and 5.3, we find that models with more heads achieve better reconstruction while producing more diverse basis vectors. This suggests that fine-grained architectures yield more expressive representations, although they may also exhibit undesired challenging behavior like feature splitting (Bricken et al., 2023) or absorption (Chanin et al., 2024).

Figure 19 visualizes this structure through UMAP projections (n_neighbors=15, min_dist=0.05, metric='cosine') of decoder weights from the first 8 heads of 32k SAEs with varying m,n configurations. The plots reveal distinct clustering patterns: for $m < n$ we observe tight base-wise clustering with weaker grouping by extension, and for $m \geq n$ extension-wise clustering is stronger.

This asymmetry suggests that pre-latent capacity requirements directly manifest in the embedding geometry - components with lower polysemanticity (extensions when m < n) exhibit greater geometric diversity. We expect symmetric behavior for reciprocal configurations (e.g., m=4,n=8 vs. m=8,n=4), merely swapping the roles of bases and extensions.

**Interpretability across sparsity regimes.** We compare KronSAE and its Matryoshka variant with TopK, Matryoshka and Switch SAE baselines across different sparsity regimes for Gemma 2 2B and Qwen 2.5 1.5B. For each SAE we follow the same pipeline as for results in Section 5.3, but use only 18 million tokens for examples collection. For KronSAE and TopK we use first 4096 features, and

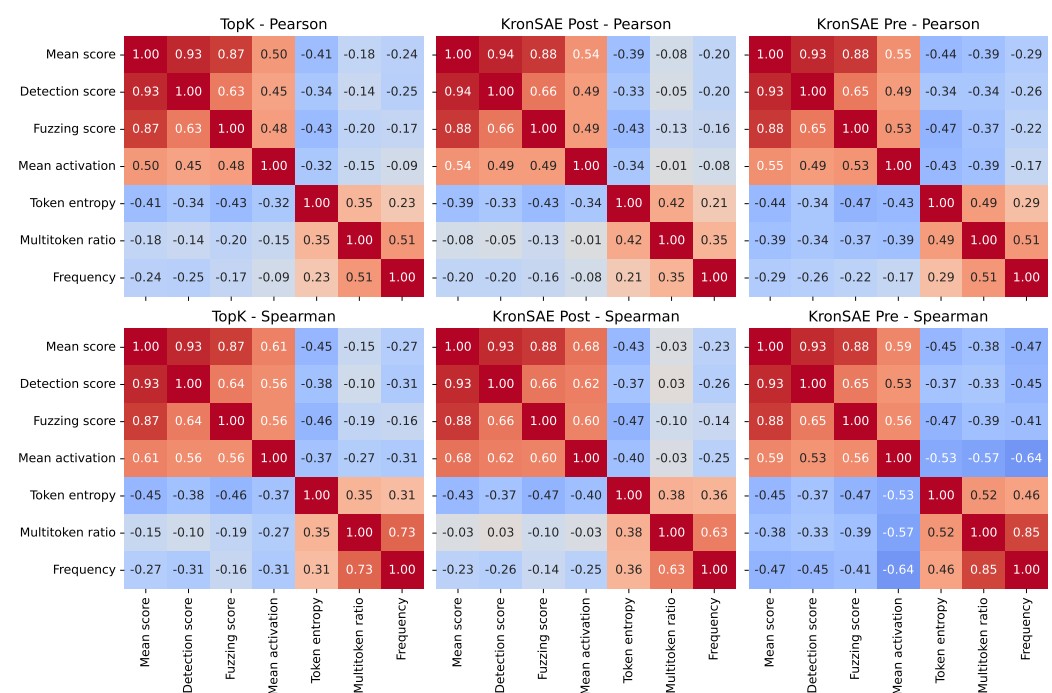

Figure 17: Correlation coefficients (Pearson and Spearman) between properties of TopK and KronSAE latents. Token entropy emerges as a strong predictor of interpretability scores, while higher mean activation and lower frequency also indicate more interpretable features.

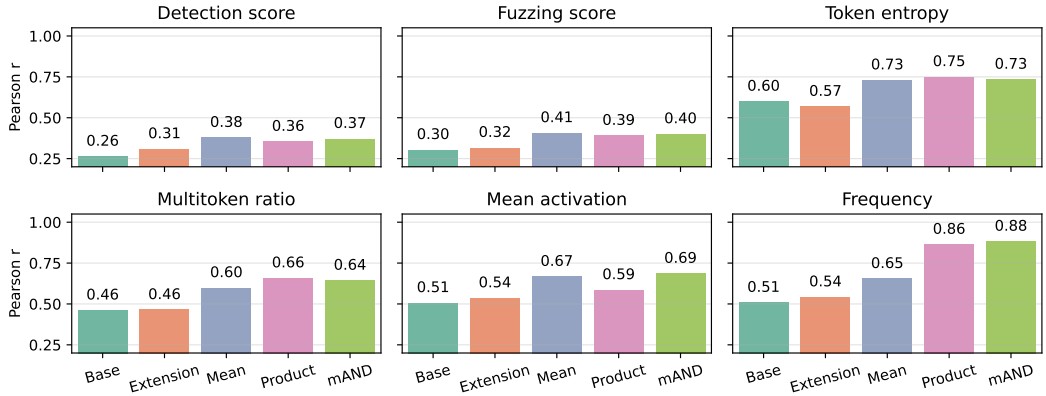

Figure 18: Correlation patterns between properties of post-latents and pre-latents.

for other models we sample 4096 features randomly. Scores for $F = 2^{16}$ and $F = 2^{15}$ are presented in Figures 20 and 21.

Results show that TopK and KronSAE are very stable across both models and sparsity regimes, while for other models their architectural and training design significantly affect the interpretability.

We attribute those differences between Gemma and Qwen to different capacity of residual stream - Gemma has hidden state size of 2304, while Qwen has only 1536 dimensions (1.5x smaller). Hovever, differences between training data and architectural choices can also be the cause.

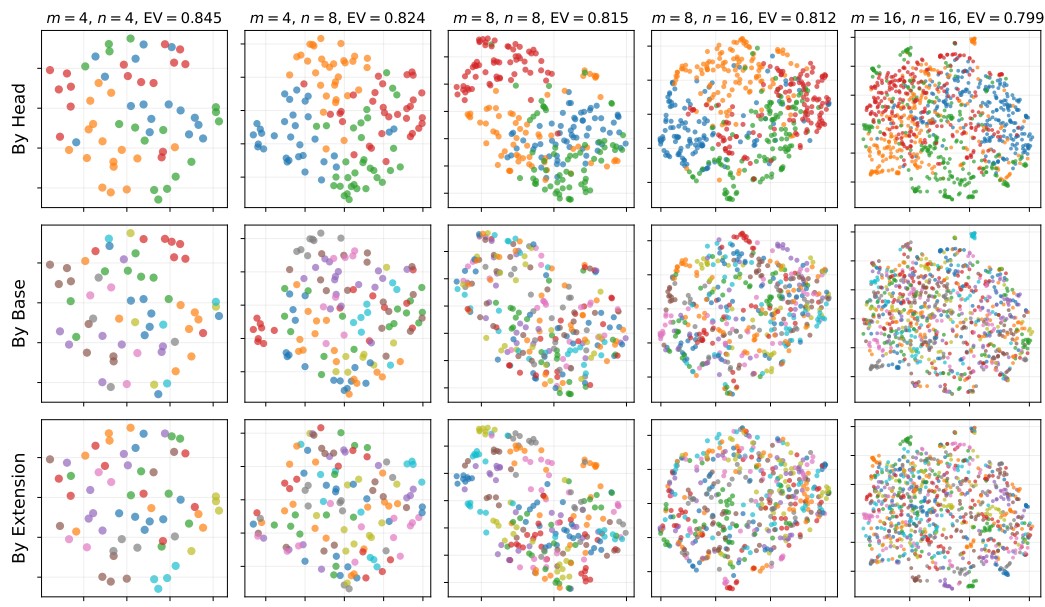

Figure 19: UMAP visualization of post-latent clustering patterns by head, base, and extension group membership. We observe tight clusters by base for $m < n$ and by extension for $m \geq n$.

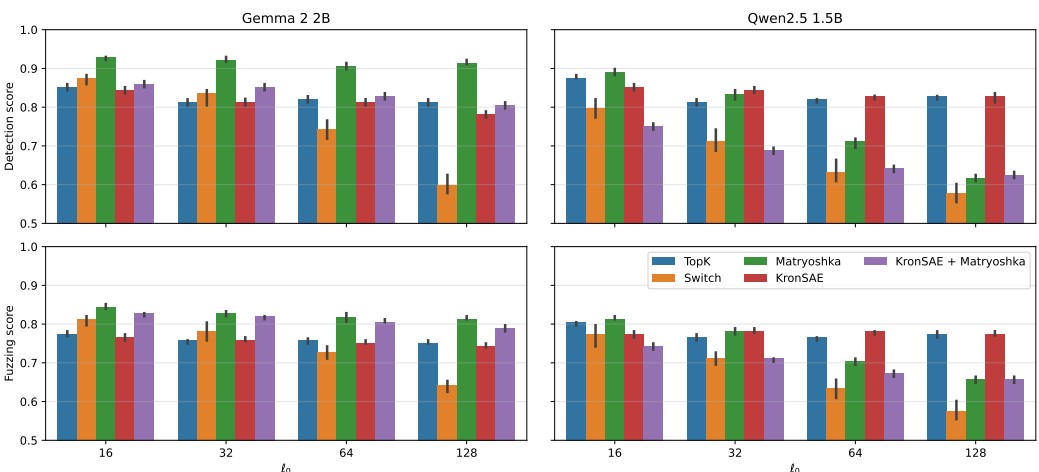

Figure 20: Interpretability scores for $F = 2^{16} = 65536$. KronSAE shows good consistency across different sparsity regimes, in some cases outperforming the TopK baseline.

**Interpretability tradeoff.** As shown in Section 4.1 KronSAE introduces tradeoff between computational efficiency, explained variance and interpretability. We can force more features to be correlated in same head by increasing the $m$ and $n$, and this will improve computational efficiency, but at the cost of reconstruction performance; however, improving the explained variance (with small $m$ and fine-grained structure of groups) goes with the cost of slight reduction in the interpretability. These tradeoffs are expected and presented across variety of SAEs (Karvonen et al., 2025).

To evaluate how KronSAE behave under different $m, n$ we compute autointerpretability scores following the same setup as described in Appendix D for Qwen-2.5 1.5B and Gemma-2 2B models with sparsity $\ell_0 \in \{16, 32, 64, 128\}$. As Figure 22 shows, setup $m = 1$ and $h = 4096$ is less interpretable than $m = 2$ and $m = 4$ with the same number of heads despite having stronger reconstruction performance.

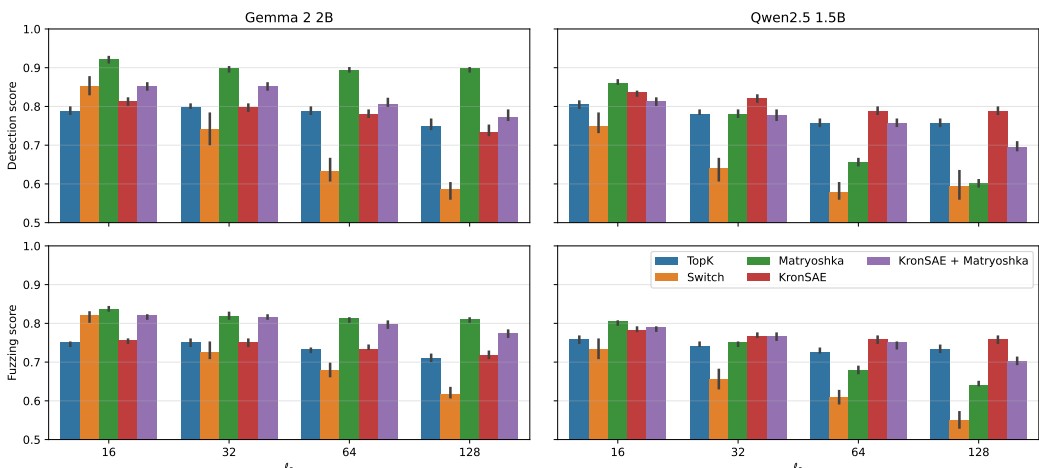

Figure 21: Interpretability scores for $F = 2^{15} = 32768$. KronSAE shows good consistency across different sparsity regimes, in some cases outperforming the TopK baseline. The result is the same as for 65k dictionary size at the Figure 20, but with slightly lower scores.

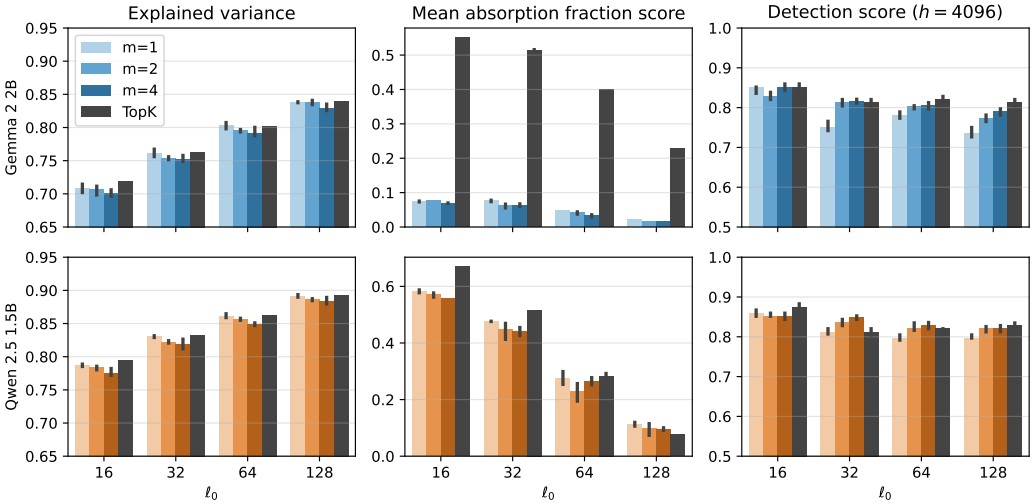

Figure 22: EV, absorption score and interpretability for $F = 2^{16}$. For constant $h$, decreasing $m$ leads to lower interepretability scores and absorption scores, but increase EV - this is expected due to increasing polysemanticity of pre-latents and entanglement of the features. See Section 5.3 for a explanation of the retrieval mechanism.

## F    KRONSAE IN TERMS OF TENSOR DIAGRAM

The proposed encoder architecture can be visualized as a tensor diagram (Figure 23). Notably, this formulation draws a connection to quantum mechanics, where $|\mathbf{f}\rangle$ represents the (unnormalized) state of two disentangled qubits described by $|\boldsymbol{p}\rangle$ and $|\boldsymbol{q}\rangle$.

If we were to sum the outputs of the encoder's heads instead of concatenating them, $|\mathbf{f}\rangle$ would correspond to a separable quantum state. This scenario can be expressed via the Schmidt decomposition:

$$|\mathbf{f}\rangle = \sum_h |\boldsymbol{p}_h\rangle \otimes_K |\boldsymbol{q}_h\rangle \,,$$

where $\otimes_K$ denotes the Kronecker product. However, preliminary experiments revealed that this alternative design results in poorer performance compared to the concatenation-based approach.

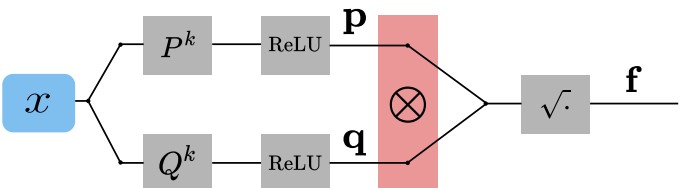

Figure 23: For a single head, the KronSAE encoder architecture separates the input $x$ into two distinct components, $p$ and $q$, via matrix multiplications $P^k$ and $Q^k$ accordingly, followed by ReLU activation. These components are then combined via the Kronecker product $p \otimes q$ and square root operation $\sqrt{\cdot}$, resulting in an output vector $\mathbf{f}$.

## G ANALYSIS OF COMPOSITIONAL STRUCTURE

Here we analyze more examples of interactions in various heads.

**Head 3.** For this head we have selected all base elements and extension 2, shown in Table 8. Extension element 2 shows moderate interpretability with clear AND-like interactions: with base 1 (semantic inheritance through shared pre-latent semantics) and base 2 (retaining only instrument-related semantics). Notable interactions occur with base 0 (acquiring medical semantics while preserving metric/number aspects) and base 3 (combining instrument semantics with necessity to yield therapy/treatment concepts). The high interpretability scores suggest potential additional encoding mechanisms beyond simple intersection, possibly related to activation magnitude, though dataset or interpretation artifacts cannot be ruled out without further validation.

| Component | Interpretation | Score |
|---|---|---|
| **Extension 2** | Scientific instruments, acronyms, and critical numerical values in technical and astronomical contexts | 0.71 |
| Base elements and their compositions with extension 2 | | |
| **Base 0** | *Interpretation:* Punctuation marks and line breaks serving as structural separators in text. | 0.66 |
| | *Composition:* Health-related metrics focusing on survival rates, life expectancy, and longevity. | 0.88 |
| **Base 1** | *Interpretation:* Numerical values, both in digit form and as spelled-out numbers, often accompanied by punctuation like decimals or commas, in contexts of measurements, statistics, or quantitative expressions. | 0.80 |
| | *Composition:* Numerical digits and decimal points within quantitative values. | 0.86 |
| **Base 2** | *Interpretation:* Nuanced actions and adverbial emphasis in descriptive contexts. | 0.71 |
| | *Composition:* Astronomical instruments and their components, such as space telescopes and their acronyms, in scientific and observational contexts. | 0.90 |
| **Base 3** | *Interpretation:* Forms of the verb "to have" indicating possession, necessity, or occurrence in diverse contexts. | 0.91 |
| | *Composition:* Antiretroviral therapy components, viral infection terms, and medical treatment terminology. | 0.87 |

Table 8: Interactions between extension 2 in head 3 and all base elements in that head.

**Head 136.** This head exhibits higher interpretability in post-latents than pre-latents. Key observations from the Table 9 include: extension 2 with base 0 narrows semantics to Illinois (likely inheriting geographical subsemantics), while interactions with bases 2-3 demonstrate complexity beyond simple intersection, often introducing additional semantics requiring deeper investigation.

| Component | Interpretation | Score |
|---|---|---|
| **Extension 2** | Hierarchical scopes, geographic references, and spatial dispersal terms | 0.78 |
| Base elements and their compositions with extension 2 | | |
| **Base 0** | *Interpretation:* Numerical decimal digits in quantitative expressions and proper nouns. | 0.79 |
| | *Composition:* The state of Illinois in diverse contexts with high significance. | 0.95 |
| **Base 1** | *Interpretation:* The number three and its various representations, including digits, Roman numerals, and related linguistic forms. | 0.84 |
| | *Composition:* Geographic place names and their linguistic variations in textual contexts. | 0.91 |
| **Base 2** | *Interpretation:* Ordinal suffixes and temporal markers in historical or chronological contexts. | 0.87 |
| | *Composition:* Terms indicating layers, degrees, or contexts of existence or operation across scientific, organizational, and conceptual domains. | 0.82 |
| **Base 3** | *Interpretation:* Question formats and topic introductions with specific terms like "What", "is", "of", "the", "Types", "About" in structured text segments. | 0.77 |
| | *Composition:* Spatial spread and occurrence of species or phenomena across environments. | 0.87 |

Table 9: Interactions between extension 2 in head 136 and all base elements in that head.

**Head 177.** Latents presented in Table 10 emonstrates more consistent AND-like behavior than Heads 3 and 136, closely matching the interaction pattern shown in Figure 2.

| Component | Interpretation | Score |
|---|---|---|
| **Extension 1** | Geographical mapping terminology and institutional names, phrases involving spatial representation and academic/organizational contexts | 0.90 |
| Base elements and their compositions with extension 1 | | |
| **Base 0** | *Interpretation:* Proper nouns, abbreviations, and specific named entities. | 0.64 |
| | *Composition:* Geographical or spatial references using the term "map". | 0.93 |
| **Base 1** | *Interpretation:* Emphasis on terms indicating feasibility and organizations. | 0.80 |
| | *Composition:* Specific organizations and societal contexts. | 0.89 |
| **Base 2** | *Interpretation:* Institutional names and academic organizations, particularly those containing "Institute" or its abbreviations, often paired with prepositions like "of" or "for" to denote specialization or affiliation. | 0.89 |
| | *Composition:* Institutional names containing "Institute" as a core term, often followed by prepositions or additional descriptors. | 0.92 |
| **Base 3** | *Interpretation:* Closure and termination processes, initiating actions. | 0.79 |
| | *Composition:* Initiating or establishing a state, direction, or foundation through action. | 0.85 |

Table 10: Interactions between extension 1 in head 177 and all base elements in that head.

## H KRONSAE SIMPLIFIED IMPLEMENTATION

```python
class KronSAE(nn.Module):
    def __init__(self, config):
        super().__init__()
        self.config = config
        _t = torch.nn.init.normal_(
                torch.empty(
                    self.config.act_size,
                    self.config.h * (self.config.m + self.config.n)
                )
            ) / math.sqrt(self.config.dict_size * 2.0)
        self.W_enc = nn.Parameter(_t)
        self.b_enc = nn.Parameter(
            torch.zeros(self.config.h * (self.config.m + self.config.n))
        )
        W_dec_v0 = einops.rearrange( # Initialize decoder weights
            _t.t().clone(), "(h mn) d -> h mn d",
            h=self.config.h, mn=self.config.m + self.config.n
        )[:, :self.config.m]
        W_dec_v1 = einops.rearrange(
            _t.t().clone(), "(h mn) d -> h mn d",
            h=self.config.h, mn=self.config.m + self.config.n
        )[:, self.config.m:]
        self.W_dec = nn.Parameter(einops.rearrange(
            W_dec_v0[..., None, :] + W_dec_v1[..., None, :, :],
            "h m n d -> (h m n) d"
        ))
        self.W_dec.data[:] = (
            self.W_dec.data / self.W_dec.data.norm(dim=-1, keepdim=True)
        )
        self.b_dec = nn.Parameter(torch.zeros(self.config.act_size))

    def encode(self, x: torch.Tensor) -> torch.Tensor:
        B, D = x.shape
        acts = F.relu(
            x @ self.W_enc + self.b_enc
        ).view(B, self.h, self.m + self.n)
        all_scores = torch.sqrt(
            acts[..., :self.config.m, None] * \
            acts[..., self.config.m:, None, :] + 1e-5
        ).view(B, -1)
        scores, indices = all_scores.topk(
            self.config.k, dim=-1, sorted=False
        )
        acts_topk = torch.zeros(
            (B, self.config.dict_size)
        ).scatter(-1, indices, scores)
        return acts_topk

    def forward(self, x):
        acts_topk = self.encode(x)
        x_rec = acts_topk @ self.W_dec + self.b_dec
        output = self.get_loss_dict(x, x_rec)
        return output

    def get_loss_dict(self, x, x_rec):
        loss = (x_rec - x.pow(2).mean()
        pt_l2 = (x_rec - x).pow(2).sum(-1).squeeze()
        var = (x - x.mean(0)).pow(2).sum(-1).squeeze()
        ev = (1 - pt_l2 / var).mean()
        return loss, ev
```

# I   USAGE OF LARGE LANGUAGE MODELS

We have used LLMs as the main tool for conducting the interpretability experiments, as described in section D, and as the instrument for language polishing and word choice.

