# OpenReview forum: "Kronecker Factorization Improves Efficiency and Interpretability of Sparse Autoencoders"
_ICLR.cc/2026/Conference — Submitted to ICLR 2026_

### Official Review · Reviewer_GyXY · 2025-10-31

**Soundness:** 2
**Presentation:** 3
**Contribution:** 3
**Rating:** 2
**Confidence:** 4

**Summary:**

The authors introduce a new variant of dictionary learning for language model interpretability. Motivated by increasing computational efficiency, the Kroneker encoder trains a 2-level hierarchy, composing multiple encoders that each operate on distinct subspaces of the embedding space.

**Strengths:**

- The architecture is substantially more parameter efficient that existing SAE architectures.
- The toy model of correlation section provides evidence of improved learning of correlations compared to TopK SAEs.

**Weaknesses:**

1a. In my view, the main novelty of this work is imposing a prior on which structures to learn: a 2-level hierarchy, for subspaces of the residual stream space. While I agree this architecture improves over compute efficiency, the paper needs a better motivation for the structural prior. Why do we expect language models to learn this specific hierarchical structure? What kind of features is a TopK SAE not able to learn, while the Kron SAE is?

1b. The analysis section provides a qualitative discussion of examples of learned feature hierarchies. Table 2 exemplifies a polysemantic base component that extends to seemingly unrelated concepts: comparative words and directional words and spiritual words. Quantifying the extent to which ground truth concept hierarches in natural language are identified but the KronSAE. Overall, the hierarchical nature of learned features remains underexplored. The Feature Absorbtion evaluation and SAEBench results provide a useful signal to compare performance of existing saes, but does not directly evaluate the recovery of feature hierarchies.

2. The hierarchical prior of KronSAEs is related to the prior of Matryoshka SAEs. I'd like to see a baseline of Matryoshka SAE scores on all evaluations.

**Questions:**

Why are heads only operating on distinct subspaces of the residual streams. What happens to LM features that are an element of the union over input spaces of multiple heads?

---

> ### Author Response · Authors · 2025-11-21
> **Response to Weakness 1a**
>
> We thank the reviewer for the thoughtful feedback. Bellow we will address weaknesses and questions point by point:
>
> ---
>
> W1a. It is natural to expect some concepts to activate only when their more general parent concepts are present. KronSAE introduces an inductive bias that supports this behavior: by composing small pre-latents, the model encourages post-latents that represent intersections of primitives and therefore active only when their “parents” are also active. Our analysis as described in section 5.3 suggest that usage of this inductive bias in KronSAE is not limited to simple binary hierarchy and is more diverse:
>
> - Example in table 2 shows pre-latent which semantics is a union of three different concepts: “Suffix “-like” for comparative descriptors, directional terms indicating geographical regions, and concepts related to spiritual or metaphysical dimensions”, each encoded via the magnitude of pre-latent activation, and the post-latents for this head are specific instantiations of these concepts. Therefore, this particular pre-latent is not just one broad concept that produces more concrete post-latents, but the union of three *distinct and identifiable* concepts at once.
> - In table 11, the extension 1 is also the some kind of union of two very broad concepts (spatial/geographical plus institutional/organizational contexts), and the post-latents have the semantics expected from the AND-like architectural constraints:
>     1. Base 2 is a very narrow one, containing primarily the “Institution” term with prepositions; its combination with extension 1 creates AND-like behaviour and produces the post-latent containing the semantics on the intersection between the pre-latents. As we hypothesize, base 2 is the subset of extension 1.
>     2. Base 3 with semantics “action initiation”, when interacting with extension 1, produces the post-latent that has the properties related to both pre-latents: from the base it tooks the initiation aspect, and from extension it took the spatial aspect, producing the “Initiating or establishing a state, direction, or foundation through action” semantics. As we hypothesize, base 3 is *not* the subset of extension 1, meaning that base 3 might be active when extension 1 is not, but their intersection is nonempty.
> - As we write in lines 414-424, other types of interaction might appear, although this is derived from the approximate interpretations of features rather than from manual hand-crafted analysis of feature behaviour (which is infeasible).
>
> Overall, these findings indicate that KronSAE is not limited to simple A → B hierarchy, and the architecture uses the activation magnitude to encode different semantics. This structural priors allow us to set the expectations of how would KronSAE features look like after training, making the interpretation easier, but not strictly bound their structure.
>
> A second, related goal of our architecture is to ensure that correlations between features (which are already presented in SAEs) have an interpretable cause. Correlated features tend to belong to the same group (head/base/extension) rather than arising spuriously across unrelated atoms. And third goal is to ease the large-scale training of SAEs for interpretability purposes, which is achieved via encoder factorization. KronSAE’s head-wise Kronecker structure addresses all these goals.
>
> Considering the features that can be learned by KronSAE, but cannot be learned by TopK, we actually see this situation the other way around: since TopK does not have any regularization on the type of the learned features, it has more broad set of potential solutions of the optimization procedure (i.e. the potential sets of learned features). It is the structural relationships that explicitly encoded in KronSAE, but not in TopK, and apparently this structure improves feature quality and balances the efficiency with performance.

---

> ### Author Response · Authors · 2025-11-21
> **Response to Weaknesses 1b, 2 and Question**
>
> W1b. We thank you for the suggestion and agree that study of relationships in learned KronSAE features is a valuable and important direction. However:
>
> 1. As described in W1a and lines 414-424, KronSAE feature structure is not strictly about nested hierarchies such as bird → sparrow or flower → rose, but rather mainly about intersection between semantics of pre-latents (the AND-like behaviour), which is different mechanism that we explicitly describe and explore in the paper. As we show in sections 5.3 and appendix G (tables 2, 9, 10 and 11), KronSAE feature structure is more complicated, and quantifying the “hierarchy recovery” will only partially explain the structure of KronSAE. For these reasons we refer to hierarchy as co-activation of two pre-latents with post-latent within one head, rather than abstraction hierarchy as expected in Matryoshka.
> 2. “Ground truth concept hierarches in natural language” is a problematic matter: databases of such ground truth hierarchies as in WordNet are not directly related to the SAE features (since features are not simply words or single-word concepts), and the “ground true LLM features that could be learned by SAE” are also yet unspecified and perhaps could not be [1-4].
>
> Therefore, it is not clear if it is suitable to quantify the “hierarchy recovery” due to point 1, and it is unclear what procedure will allow proper quantification due to point 2. If you have a recommendation for a specific experiment, we would be glad to run it and include the results in the revision.
>
> We will add clarifications in a revised version of our work to make it more easy to follow.
>
> ---
>
> W2. We have ran additional experiments with Matryoshka and Switch SAEs as baselines on Qwen-2.5 1.5B and Gemma-2 2B models under the iso-FLOPS setting as described in our paper. The following results show the performance of the baselines and their KronSAE variants on 65k dictionary size and with 500M tokens as reference budget, all SAEs aligned with TopK SAE FLOPS:
>
> https://postimg.cc/0MBrnDM1
>
> The next results on feature absorption suggest that applying KronSAE encoder improves absorption scores, especially combined with Matryoshka dictionary learning:
>
> https://postimg.cc/3yZvXkF8
>
> And the following results are for interpretability scores:
>
> 32k dictionary: https://postimg.cc/GBB6f436
>
> 65k dictionary: https://postimg.cc/WDJyL9xx
>
> Additionally, we have included the Matryoshka SAE baseline to the budget-wise evaluation:
>
> For Qwen2.5 1.5B: https://postimg.cc/1fjLcGX3
>
> For Gemma 2 2B: https://postimg.cc/HJ4ZKXY4
>
> We will include all these results in the paper with detailed description of hyperparameters.
>
> We note that KronSAE is more than another SAE variant: its head-wise Kronecker-factorised encoder and mAND operator are architectural primitives that can be applied to any SAE with a dense encoder (not TopK only). All these approaches are orthogonal to each other: one could build the Switch SAE with Kronecker-factorized experts (as in KronSAE) trained with Matryoshka-like loss, combined with JumpReLU activation function.
>
> ---
>
> Q1. We work within the framework of the Linear Representation Hypothesis (LRH) and therefore treat features as one-dimensional directions in the resudual stream space [5, 6]. All heads receive the same input and read information from the same space, but each head reads that input through its own projections (the $P_k, Q_k$ matrices) and therefore process shared information differently. So the difference between heads is in how they read and combine the same input. Any feature that lies in the union of a head’s receptive directions is simply a feature that at least one head can identify; in practice, KronSAE’s encourage correlated concepts to be localized inside a single head, while TopK SAE has no structural bias in what features it would learn.
>
> ---
>
> Thank you again for your insightful comments. We hope that our answer address your concerns and you will find our clarifications and additional rationales helpful. Let us know if you have further questions or concerns.
>
> [1] Sparse Autoencoders Do Not Find Canonical Units of Analysis, Leask et al., 2025
>
> [2] Sparse Autoencoders Trained on the Same Data Learn Different Features, G. Paulo, N. Belrose, 2025
>
> [3] Archetypal SAE: Adaptive and Stable Dictionary Learning for Concept
> Extraction in Large Vision Models, Fel et al., 2025
>
> [4] Position: Mechanistic Interpretability Should Prioritize Feature Consistency in SAEs, Song et al., 2025
>
> [5] Toy Models of Superposition, Elhage et al., 2022
>
> [6] Emergent linear representations in world models of self-supervised sequence models, Nanda et al., 2023

---

### Official Review · Reviewer_dNqs · 2025-11-01

**Soundness:** 2
**Presentation:** 2
**Contribution:** 2
**Rating:** 2
**Confidence:** 3

**Summary:**

This paper proposes a new sparse autoencoder (SAE) architecture, KronSAE, where the SAE's encoder consists of multiple Kronecker-factored blocks. They present iso-FLOP comparisons between KronSAEs and TopK SAEs in terms of reconstruction, feature absorption, and interpretability.

**Strengths:**

1. The KronSAE architecture is clearly described.
2. In some sections, the authors sweep over certain key hyperparameters (F and m) to understand their effect.
3. I found the discussion of the relationship between pre- and post-latent interpretations interesting.

**Weaknesses:**

Overall I would like to see more systematic reporting of results:
1. Multiple models. KronSAEs are trained on three models, but most results are only reported for one model. This makes it difficult to tell if results are cherry-picked.
2. Consistent values of m and F. The reconstruction performance results are shown for multiple values of m, with m=1 being the best. But later, only m=4 is shown for interpretability results. Since, in general, we should expect there to be a trade-off between reconstruction and sparsity (which is typically correlated with interpretability), I worry that there is a tradeoff to KronSAEs which is not being shown here.
3. Multiple sparsities. Figures 3 and 4 sweep over multiple sparsities; ideally all of the plots would do the same.
Overall, I would find it much easier to understand the results if the plots in this paper were replaced with line plots where the x-axis was sparsity, and there were multiple lines corresponding to different values of m. There should be one such plot for each model (though some of them can be reported in the appendix).

Other notes:
1. KronSAEs are only compared to TopK SAEs. As the authors note, Matryoshka SAEs are an idea in a somewhat similar vein. So it would be better if KronSAEs were also compared at least against Matryoshka SAEs as well.
2. Reconstruction results are best when m=1, but this is also the case when KronSAEs are most similar to the standard architecture; the reconstruction hit is more substantial for m=4,8.

If these concerned are addressed, I could see myself raising the score to as high as 6.

**Questions:**

1. Is there a reason we would want to localize correlated features to the same head? The paper writes about this as if it's a desirable property, but it's not clear to me why it matters.
2. This paper makes the choice to study KronSAEs in a resource-constrained setting, i.e. where SAEs are trained for <=1B tokens. I'm curious if you have any sense of what the results look like when training for longer.

---

> ### Author Response · Authors · 2025-11-23
> **Response to Review**
>
> We thank the reviewer for the constructive and detailed feedback. We appreciate that you found (1) the KronSAE architecture clearly described, (2) clear hyperparameters sweep, and (3) the discussion part interesting. Below we address raised concerns.
>
> ---
>
> W1. We have run additional experiments on Gemma and Qwen so there are more consistency. First, we have ran the iso-flops comparison with TopK also on Gemma model:
>
> https://postimg.cc/FfGx9Wqn
>
> Below is the comparison with the baselines - TopK, Switch SAE and Matryoshka SAE, and for each baseline we have also measured the performance of its KronSAE variant to show that KronSAE might be combined with already existing architectures:
>
> https://postimg.cc/0MBrnDM1
>
> The next result is for absorption score comparing TopK, Matryoshka and KronSAE:
>
> https://postimg.cc/3yZvXkF8
>
> The following results are for interpretability scores:
>
> 32k dictionary: https://postimg.cc/GBB6f436
>
> 65k dictionary: https://postimg.cc/WDJyL9xx
>
> All these results suggest that KronSAE effectively balances the reconstruction performance, interpretability of latents and parameter efficiency. We will add them in the revised version with all necessary details to ensure reproducibility.
>
> W2. Our KronSAE gives tradeoff between FLOPs, number of parameters and explained variance as we report in the paper. We can force more features to be correlated in same head by increasing the $m$ and $n$; however, this requires pre-latents to be polysemous and therefore encode multiple semantics (as we write in section 5.3 about retrieval mechanism), also increasing the entanglement between latent activations. Therefore, from a qualitative point of view the main difference is the correlational structure in the head and the degree of polysemanticity of pre-latents.
>
> In our experiments, varying $m, n$  produced minimal changes in interpretability scores (except for m16n16, but we believe this is stochastic outlier since m32n32 has the same result as m8n8):
>
> https://postimg.cc/bZBqzBrs
>
> However, the following result for 65k dictionary shows that m4n4 setup is slightly better than m2n4 and m2n8, although difference is also small:
>
> https://postimg.cc/FkV9NFkQ
>
> Please also refer to results in our answer to W1.
>
> W3. We thank you for this valuable suggestion. We ran multiple evaluations for different sparsity regimes on Gemma and Qwen models. Please refer to our answer to the W1.
>
> ---
>
> N1. We provide additional experiments with baselines and their KronSAE variants, please see our answer to W1.
>
> N2. You are correct, and this is the manifestation of the tradeoff between parameter efficiency and reconstruction performance.
>
> ---
>
> Q1. The presence of correlated features in SAEs suggests that a dictionary can be used more compactly and it is desirable for those correlations to have an interpretable cause. KronSAE introduces such a structural inductive bias in two complementary ways: (1) a two-level hierarchy that forces the post-latents to co-occur with their pre-latents, and (2) head-wise grouping that produces adjacent groups of post-latents so that their co-occurrence has an interpretable cause (i.e. the same parent pre-latents). We do not see the correlations as a problem that should be solved, but our approach shows that this parameterization and utilization of correlational structure might be beneficial.
>
> Q2. Because KronSAE reduces the encoder parameter count, it will reach the plateau earlier and the plateau level would be lower. Conversely, with longer training more expressive models would continue to improve the reconstruction performance and the plateau level would be higher. This behaviour is expected and explicitly discussed in the Limitations section. Another observation is that given larger training budget, the gap between KronSAE and TopK SAE would be smaller on larger dictionary sizes. To illustrate these dynamics we add experiments on Qwen with 2B tokens as the reference budget:
>
> https://postimg.cc/MfRRzBqt
>
> ---
>
> We hope that our answer addresses your concerns. If you have further questions, suggestions or concerns, please let us know. Thank you for your comments that have improved our paper!

---

> > ### Comment · Reviewer_dNqs · 2025-11-24
> >
> > Thank you for these additional results.
> >
> > Can you clarify which dictionary size is used for the first 2nd and 3rd links in your response?
> >
> > Also, can you please provide interpretability results for the same m=1 KronSAEs that you find yield reconstruction on par with the baseline SAEs? To be explicit, I'm worried that by choosing m=1 you can get good reconstruction and by choosing m > 1 you can get good interpretability, but that there is a trade-off between these that is being hidden.

---

> > > ### Author Response · Authors · 2025-11-25
> > > **Response to Official Comment by Reviewer dNqs**
> > >
> > > > Can you clarify which dictionary size is used for the first 2nd and 3rd links in your response?
> > >
> > > The dictionary size is 65536 on the second and third plots, and the first plot has different and emphasized dictionary sizes (32k, 65k and 131k).
> > >
> > > > I'm worried that by choosing m=1 you can get good reconstruction and by choosing m > 1 you can get good interpretability, but that there is a trade-off between these that is being hidden.
> > >
> > > Thank you for clarifying your concern further. There is indeed a tradeoff, and we agree that it is necessary to mention it. We did not intend to hide it but rather did not examine it since $m=1$ setup suffers from efficiency and reconstruction performance tradeoff - although it improves EV, it decreases speed and increases the amount of trainable parameters, which is not desirable since the EV improvement might be negligible, but the interpretability cost and training effort overhead might be substantial; some setups with $m=1$ have even more trainable parameters than the TopK baseline. It is some kind of extreme case of our architecture.
> > >
> > > To enhance the systematicity of presentation, we computed the autointerpretability scores for already trained KronSAE variants with dictionary size 65k and $h=4096$ and plot it with our core metrics so the reader could better capture the tradeoff:
> > >
> > > https://postimg.cc/bSYR4BCQ
> > >
> > > This result suggest that decreasing $m$ and keeping the same amount of heads lowers the interpretability - this is expected due to increasing polysemanticity of pre-latents and entanglement of the features (we refer to our answer to your W2). Increasing the number of heads (e.g. choosing $m=1, n=8$, or $m=1, n=4$, etc.) would worsen the computational performance. The tradeoff between explained variance and interpretability is also present in other SAE models [1], so this is expected behaviour.
> > >
> > > We will explicitly describe this tradeoff for KronSAE in a paper.
> > >
> > > ---
> > >
> > > Additionally, we have included the Matryoshka SAE baseline to the budget-wise evaluation:
> > >
> > > For Qwen2.5 1.5B: https://postimg.cc/1fjLcGX3
> > >
> > > For Gemma 2 2B: https://postimg.cc/HJ4ZKXY4
> > >
> > > ---
> > >
> > > We hope that you find our answer satisfactorily.
> > >
> > > [1] https://www.neuronpedia.org/sae-bench

---

### Official Review · Reviewer_eE5v · 2025-11-07

**Soundness:** 4
**Presentation:** 3
**Contribution:** 2
**Rating:** 6
**Confidence:** 5

**Summary:**

This paper introduces a new SAE variant with a structured encoder, called KronSAE. Instead of using a since encoder matrix, this work uses a sum of smaller Kronecker factored matrices. While it doesn't improve training speed or final reconstruction, it improves over the baseline in two domains:
- Lower parameter count and all the subsequent advantages.
- A clustering-based (or AND-based) prior for feature extraction, reducing feature splitting.

This leads to extracted features that are more useful for interpretation over the baseline.

**Strengths:**

Introducing structured priors into SAEs is useful. This offers both regularization for the training process but also helps interpretability of latents. I think this paper hence makes meaningful progress in an important domain.

Despite my later comments nit-picking sections, the overall presentation is sound. The text is clearly structured and the story makes sense.

The presented experiments are thorough and most my questions were answered immediately.

**Weaknesses:**

--- **Weaknesses** ---

The correlation experiment seems heavily favoured towards your approach since this is precisely what the Kronecker structure relies on for extraction. It's nice to see the KronSAE succeeds but I'm fairly certain there's an equally contrived experiment where TopK will find ground truth structure much better than KronSAEs. Actually, looking at Figure 9, it seems that this correlation plot is basically the same regardless of the original patterns, which further undermines this qualitative experiment.

Not sure why mAND is featured so prominently, if the simpler setup of simply doing $u*v$ is only 1% worse, why not just use that? If I'm not mistaken, this is extremely akin to other efficient matrix factorizations like Butterfly/Monarch matrices. The use of these efficient parameterizations is not new in deep learning [1] and a more thorough literature discussion mentioning this would be useful.

[1]: https://arxiv.org/abs/2204.00595

--- **Improvements** ---


The notation was also a bit confusing to me because bold characters are often used to indicate something being a vector of the same matrix (especially if indexed), e.g. $p^k$ is row $k$ of $P$. While it's the text makes it clear, I recommend changing notation a bit. On a related note, why does table 1 use $u$ and $v$ rather than $p$ and $q$, they're the same thing right?

While satisfactory, the explanation in Section 3 could be easily improved by adding a diagram. I saw there is one in appendix F but it uses strange notation (and doesn't include the top-k). The diagram should be (in einops notation), einsum(x, x, Q, P, "... xtop, ... xbot, xtop q h, xbot p h -> (p q h)"). The Kronecker product is implicit.

Figure is quite hard to read. perhaps there's a nicer way to present this? Also, since you introduced a new metric it seem useful to walk the reader through what they're seeing here. It took some time to figure that out myself.

In general, many of the figures are too cluttered to read easily. I suggest extracting the important bits and moving the full figure to the appendix or something.

**Questions:**

*"We had not observed any notable differences in feature geometry between TopK and our SAEs"*\
How did you measure or observe this?

If mAND just the RELU variant but with a square root? What's the reason this was preferred outside the slightly higher reconstruction scores?

You mention the KronSAE is unstable w.r.t n, m and h. What happens qualitatively to the features? Any idea why it fails for specific setups? Answering these questions would provide some insight into the representation structure.

Do you intend to share the code?

---

> ### Author Response · Authors · 2025-11-23
> **Response to Weaknesses**
>
> We sincerely thank you for the careful and constructive evaluation of our work, especially for highlighting the usefulness of structured priors, the clarity of our exposition, and the thoroughness of the experiments. In the following, we address the concerns you raised in your review:
>
> W1. Since KronSAE design was also motivated by the existence of correlations between features and designed to concentrate correlated features within groups (heads), this particular experiment is designed to verify that KronSAE successfully learns this correlational structure, comparing it to the unconstrained architecture without architectural priors - TopK SAE. In our work, we explicitly refer to “ground truth” correlations only in the toy example experiments where we know it since we set it by ourselves, while in case of natural language we only quantify the presence of inter-group and between-group correlations. Thank you for raising this point, we will make it clearer in the revised version. The TopK is indeed might better recover the ground truth structure if it has another form (and if there exist any), but this is out of scope of our current work.
>
> W2. To emphasize the distinction between Kronecker-factorized encoder (a mathematical operation on vectors in our case) and logical operation AND that emit the signal only when the both operands are active, we treat these two design choices as separate elements, although they positively interfere with each other in our architecture. Hence, the second choice (mAND) is a subject for ablation study and therefore we reserve a space in the text for it and mention it repeatedly. Considering the *why* aspect, mAND's geometric mean ($\sqrt{uv}$) preserves activation magnitude stability during training, which is also useful for interpretability, whereas raw product ($uv$) might cause activation explosion when both pre-latents increase. Due to the above reasons accompanied with better results on ablation, we stick with the variant reported in paper.
>
> Butterfly, Monarch, and related efficient matrix factorizations focus on computational structure for matrix multiplications and properties of linear transformations. In contrast, KronSAE is a reparameterization of encoding mechanism that leverages the Kronecker product between pre-latent vectors $\mathbf{p} \in \mathbb{R}^m, \mathbf{q} \in \mathbb{R}^n$, obtained via small dense projections of residual stream input, creating the AND-like interaction structure. The approach therefore introduces a structural inductive bias at the representation-composition level rather than an alternative matrix multiplication primitive. While they are of slightly different purpose, we thank you for suggesting these - we will include the discussion of them as of possible approach to induce specific properties of encoding mechanism, and as alternative approach to improve training and inference efficiency.

---

> ### Author Response · Authors · 2025-11-23
> **Response to Improvements**
>
> I1. Thank you for pointing at this, we will improve the notation in a revised version. The  $u, v$ are used in a different context - they are used to describe the mAND operator and comparison with other variants, while $\mathbf{p}, \mathbf{q}$ are obtained after ReLU and are used to describe the KronSAE architecture.
>
> I2. We believe that there is a possibility of slight misunderstanding of our approach and would like to clarify it to avoid potential confusions. The input tensor has the dimension $d$ (hidden dimension of the model) for both $P$ and $Q$ matrices, hence $\text{xtop} = \text{xbot} = d$. The $\mathbf{p}$ and $\mathbf{q}$ are vectors of dimensions $m$ and $n$ respectively. From a mathematical point of view, the $P$ and $Q$ matrices correspond to a single head and do not have the head dimension; from a computational point of view, we are not dividing the encoder matrix into submatrices but rather use a single projection before the Kronecker product.
>
> We agree with your suggestion that diagram would improve the presentation and your variant seems to be equivalent to our approach, but in current form it seems to not represent mathematical structure nor computational; instead, we think of representing the flow within a single head, i.e. (1) dividing the computational process into two pathways, one for $P$ and one for $Q$, applying the ReLU to each other, producing the $\mathbf{p}$ and $\mathbf{q}$, (2) applying the Kronecker product between these vectors followed by the square root and (3) TopK operator applied to the resulting activations vector. This will clearly describe the flow, and we plan to include this diagram in a revised version of a manuscript (improving the diagram in Figure 16).
>
> I3. Could you please clarify which Figure you refer to? We would be glad to improve the presentation, but we are unable to derive its number from the context.
>
> I4. Thanks for highlighting it.
>
> Overall, we thank you for these suggestions and will try our best to incorporate them into revised version of our paper along with other improvements, balancing the clarity and density of presentation.

---

> ### Author Response · Authors · 2025-11-23
> **Response to Questions**
>
> Q1. We have not dive deep into geometry analysis and examined feature geometry only by analyzing the cosine similarity structure and spatial distribution of decoder weight vectors, also using dimensionality reduction methods to capture the clusterization structure (as reported in Appendix E, paragraph Basis geometry). While we observed expected clustering by head/base/extension membership within KronSAE (Figure 15), as we report in lines from 429, we have no other evidence of differences between TopK and KronSAE geometric properties. If you have suggestions or experiment setups that you interested in we will glad to run it!
>
> Q2. Please see explanation in W2. We will also slightly improve notation by directly incorporating the ReLU into mAND design, since now it seems as a separate element.
>
> Q3.  While we do report that choice of $h, m, n$ affects both the performance and feature structure, we note that we do not explicitly consider it as instability - rather, this is an expected outcome of KronSAE tradeoffs (parameter efficiency, pre-latent polysemanticity and reconstruction performance) that has understandable cause and simple solution (as described in lines 164-166 and in appendix A.5).
>
> Answering your question, we can increase the head capacity by increasing the $m$ and $n$; however, this requires pre-latents to be polysemous and therefore encode multiple semantics (as we write in section 5.3 about retrieval mechanism): some pre-latent must encode multiple semantics and the corresponding pre-latents retrieve this semantics via AND-like mechanism. Therefore, from a qualitative point of view the main difference is the correlational structure in the head and the degree of polysemanticity of pre-latents, and we attribute failure modes primarily to the insufficient capacity of the pre-latents.
>
> In our experiments, varying $m, n$  produced minimal changes in interpretability scores (except for m16n16, but we believe this is stochastic outlier since m32n32 has the same result as m8n8):
>
> https://postimg.cc/bZBqzBrs
>
> However, the following result for 65k dictionary shows that m4n4 setup is slightly better than m2n4 and m2n8, although difference is also small:
>
> https://postimg.cc/FkV9NFkQ
>
> Geometrically, changing the $m$ and $n$ produces the clusterization structure described in section 5.3, paragraph Geometry of post-latents, and in appendix E, paragraph Basis geometry.
>
> If you have suggestions for additional analysis, we would be also glad to conduct it.
>
> Q4. The code has been attached in the supplementary materials. We intend to make it publicly available once it finalized and polished upon acceptance.
>
> ---
>
> Thank you again for your valuable suggestions and comments that improved our paper! Please let us know if you have further questions and suggestions, or have additional concerns to address.

---

> > ### Comment · Reviewer_eE5v · 2025-11-23
> >
> > Thanks for the detailed response, some replies or remaining remarks.
> >
> > W1: Thank you. What about my concern that Figure 9 roughly contains the same patters regardless of the ground truth? If the groups are always finding correlated structure, regardless of the underlying correlation, then that's a big weakness. This is currently my biggest blocker for not giving a higher score. In other words, the accuracy only shows that the encoder can "work around" the structural constraints, not that the structural constraints work well, and it's this (non cherry-picked) evidence I'm looking for.
> >
> > W2: What's the difference between an encoding mechanism and a matrix multiplication? I'm not sure this is a productive distinction. An encoder is a matrix multiplication between a (structured) matrix and activations, it's often followed by a non-linearity but that doesn't matter. Both your approach and the structured matrices induce a prior.
> >
> > I1: Ah I see, thanks.
> >
> > I2: I'm a bit confused by the author's comment. I understand `xtop = xbot = d`, they just need to have different indices in the `einsum`, otherwise they're element-wise multiplied. If my `einsum` is incorrect, what is the correct one (ignoring top-k).
> >
> > I3: My apologies, I was referring to Figure 7 but Figure 2 is also hard to read.
> >
> > Q1: I have no immediate suggestion for an experiment, it's a difficult problem. The statement seems hard to verify or falsify, hence the question. Minor suggestion is to remove it if it's non-sequitur.
> >
> > Q4: Thanks

---

> > > ### Author Response · Authors · 2025-11-30
> > > **Response to Weakness 2, Improvements 2 and 3**
> > >
> > > W2. It is true that both our approach and structured matrices induce a prior, but they do it differently. The matrix multiplication followed by a nonlinearity is a one way out of many to parameterize the encoding mechanism (as in TopK SAE or dense MLP layers). Our encoder design is not a simple linear projection with specific properties, but a nonlinear operation that includes ReLU followed by the Kronecker product; there is no matrix $W$ that satisfies $W \mathbf{x} = \text{ReLU}(P \mathbf{x}) \otimes \text{ReLU}(Q \mathbf{x})$. It is very different from what structured matrices allow to do.
> > >
> > > There are other instances of SAEs with modified encoder: in Switch SAE encoder divided into many experts with a routing mechanism, Gated SAEs [8] have gating mechanism (Hadamard product) along with the dense input projections. Our method and the mentioned methods replace the matrix multiplication with more sophisticated encoding scheme. The learnable matrices in our architecture and the mentioned architectures have no structure - it is the mechanism of encoding that has it. There are many other potential parameterizations of the encoder and we study the specific one - KronSAE — that has clear intuition directly embedded in its design.
> > >
> > > Still, structured matrices you mention are indeed related to our work, but they just achieve the same purpose in different way (imposing a inductive bias and making computations more efficient); to the best of our knowledge, structured matrices have never been utilized for interpretability purposes yet, and it might be an interesting and fruitful direction of future work.
> > >
> > > I2. In our previous answer we have written the detailed explanation of architectural scheme to avoid potential confusions and improve mutual understanding. The correct pseudocode should be written as follows:
> > >
> > > ```python
> > > p = relu(einsum(x, P, "... d, h m d -> ... h m"))
> > > q = relu(einsum(x, Q, "... d, h n d -> ... h n"))
> > > z = sqrt(einsum(p, q, "... h m, ... h n -> ... h m n"))
> > > f = topk(rearrange(z, "... h m n -> ... (h m n)"), k=k)
> > > ```
> > >
> > > The actual PyTorch implementation has slightly different flow and instead of multiplying the hidden state by matrices P and Q, it leverages single encoder projection with slicing afterwards, and then applies outer product followed by the square root and flattening. The simplified implementation is provided in appendix H.
> > >
> > > I3. We will try our best to make these plots more understandable by providing additional context and guidance, thank you for highlighting this.
> > >
> > > [1] Sparse Autoencoders Trained on the Same Data Learn Different Features, G. Paulo, N. Belrose, 2025
> > >
> > > [2] Mechanistic Permutability: Match Features Across Layers, N. Balagansky, I. Maksimov, D. Gavrilov, 2025
> > >
> > > [3] Analyze Feature Flow to Enhance Interpretation and Steering in Language Models, Laptev et al., 2025
> > >
> > > [4] Archetypal SAE: Adaptive and Stable Dictionary Learning for Concept Extraction in Large Vision Models, Fel et al., 2025
> > >
> > > [5] The effective rank: A measure of effective dimensionality, O. Roy, M. Vetterli, 2007
> > >
> > > [6] Sparse Autoencoders Do Not Find Canonical Units of Analysis, Leask et al., 2025
> > >
> > > [7] Position: Mechanistic Interpretability Should Prioritize Feature Consistency in SAEs, Song et al., 2025
> > >
> > > [8] Improving Dictionary Learning with Gated Sparse Autoencoders**,** Rajamanoharan et al., 2024

---

> ### Author Response · Authors · 2025-11-30
> **Response to Weakness 1**
>
> Thank you for raising further remarks.
>
> W1. We understand your concern. To resolve it, we provide the arguments below.
>
> Due to permutation invariance in SAEs feature sets, our visualizations lacked some kind of feature ordering rule. Hence raw features in KronSAE appear to look almost randomly, making these plots appear to be almost the same.
>
> To ensure clarity, we have conducted the following experiment:
>
> 1. We train autoencoder and SAEs as usual, using data with specific correlation structure, which imitates natural language;
> 2. After training, for each SAE we match its feature directions in the hidden state space (stored in decoder weights) with the autoencoder feature directions (stored in its encoder) and obtain one-to-one mapping between feature sets as done in various SAE studies [1, 2, 3, 4] that target the lack of feature ordering constraints in SAEs;
> 3. We then build the pairwise cosine similarity matrices between feature directions within autoencoder and all SAEs and compute RV score between the resulting matrix and the ground truth covariance matrix.
>
> For eight different covariance matrices, we obtain the following result:
>
> https://postimg.cc/ctskrPG8
>
> It shows that patterns learned in KronSAE resemble the underlying ground truth structure and are not random.
>
> To validate that KronSAE depends on the underlying covariance matrix and is not agnostic to it, we analyze how variable the matrices for KronSAE are. We compute two quantities that describe the statistical information contained in the matrix:
>
> - Effective rank of the covariance matrix (for SAEs it computed for the resulting pairwise cosine similarity matrices) that describe the intrinsic dimensionality of the data. We first compute the distribution of matrix singular values by dividing them by their sum, then its Shannon entropy is computed and exponentiated [5].
> - Mean correlation as the average of off-diagonal values of the resulting matrix.
>
> Then, across various setups, we estimate the average value and the standard deviation between properties of ground truth matrix and SAE matrices:
>
> | Model | RV coefficient | Effective rank | Mean correlation | Rank diff. | Mean corr. diff. |
> | --- | --- | --- | --- | --- | --- |
> | AE | 0.599 $\pm$ 0.169 | 63.3 $\pm$ 0.3 | 0.126 $\pm$ 0.043 | 47.9 | 0.179 |
> | TopK | 0.142 $\pm$ 0.023 | 58.5 $\pm$ 1.2 | 0.083 $\pm$ 0.006 | 43.2 | 0.222 |
> | KronSAE h2m4n32 | 0.357 $\pm$ 0.154 | 53.8 $\pm$ 3.4 | 0.079 $\pm$ 0.014 | 38.5 | 0.226 |
> | KronSAE h4m2n32 | 0.267 $\pm$ 0.082 | 46.0 $\pm$ 7.8 | 0.103 $\pm$ 0.034 | 30.7 | 0.202 |
> | KronSAE h4m8n8 | 0.272 $\pm$ 0.087 | 48.8 $\pm$ 5.0 | 0.091 $\pm$ 0.021 | 33.5 | 0.214 |
> | KronSAE h8m2n16 | 0.205 $\pm$ 0.044 | 55.9 $\pm$ 1.7 | 0.073 $\pm$ 0.009 | 40.6 | 0.232 |
>
> All configurations of KronSAE shows markedly stronger variability than TopK SAE and stronger average agreement with the effective rank of the underlying covariance matrix, and this clear and pronounced difference directly answers “No” to the question “Does KronSAE learn patterns that are indifferent to the underlying covariance in the dataset?”
>
> Please note that we do not claim that KronSAE reconstruct the underlying covariation directly in its heads and pre-latent groups. Kronecker factorization forces correlations to exist and be located in the heads and pre-latent groups, and this improves the correlation reconstruction and overall SAE feature quality as we show in our experiments. But this not necessarily means that learned feature sets and hence correlation structure would directly resemble the ground-truth sets of features due to unsupervised learning of SAEs and very broad solution space - every SAE finds its own feature set, and it was experimentally shown that even the same SAE can find different features in different runs [1, 4, 6, 7].

---

### Official Review · Reviewer_Hh9i · 2025-11-07

**Soundness:** 4
**Presentation:** 4
**Contribution:** 4
**Rating:** 10
**Confidence:** 4

**Summary:**

The paper introduces the KronSAE Sparse AutoEncoder (SAE) architecture, which decomposes latent spaces using Kronecker-factorization across multiple heads. Theoretically, KronSAE improves computational efficiency relative to prior SAE approaches; and empirically, authors show across a wide range of experiments and ablations that KronSAE substantially reduces feature absorption, better captures compositional structure in latent spaces, and improves interpretability of learned features.

**Strengths:**

The work introduces an innovative SAE architecture, KronSAE, that makes substantive improvements over prior SAEs in several dimensions (efficiency, feature absorption, compositionality, and interpretability), and authors clearly demonstrates these improvements empirically across comprehensive experiments. Each of these improvements presents real value to the community in their own right; and taken together, KronSAE represents a clear and significant contribution to the SAE literature.

**Weaknesses:**

I do not see any particularly substantive weaknesses in terms of the technical contributions and empirical work presented in the paper.

My one concern is that the paper fails to cite any works from the very closely-related research area of tensor product representation (TPR). TPR, first introduced in 1990 [1], has long studied how to encode compositional representations in dense embedding vectors via tensor products (a generalization of the Kronecker product), with more recent works leveraging TPR to interpret compositional representations in LLMs [2-5].
- Note that I do not believe this significantly erodes the novelty of the paper -- to my knowledge, there is no work applying TPR directly to SAEs, which I understand as being the primary contribution in this work -- but it is important to acknowledge this large body of highly relevant work and compare it with KronSAE. For instance, [3-4] also leverage dictionary learning with TPR to interpret neural representations; but KronSAE requires less advance knowledge of specific role/filler features to look for, and is more general in terms of where it can be applied to interpret model representations (e.g., [4] requires approximating the mapping from input token embeddings (fillers) all the way to the layer whose activations are being reconstructed; whereas SAEs, including KronSAE, can be easily applied to any layer without having to approximate the full model up to that layer).

[1] Smolensky, P. (1990). Tensor product variable binding and the representation of symbolic structures in connectionist systems. Artificial intelligence, 46(1-2), 159-216.
[2] Smolensky, P., McCoy, R., Fernandez, R., Goldrick, M., & Gao, J. (2022). Neurocompositional computing: From the central paradox of cognition to a new generation of ai systems. AI Magazine, 43(3), 308-322.
[3] McCoy, R. T., Linzen, T., Dunbar, E., & Smolensky, P. (2019). RNNs implicitly implement tensor-product representations. In International Conference on Learning Representations.
[4] Soulos, P., McCoy, R. T., Linzen, T., & Smolensky, P. (2020, November). Discovering the compositional structure of vector representations with role learning networks. In Proceedings of the Third BlackboxNLP Workshop on Analyzing and Interpreting Neural Networks for NLP (pp. 238-254).
[5] Smolensky, P., Fernandez, R., Zhou, Z. H., Opper, M., & Gao, J. (2024). Mechanisms of symbol processing for in-context learning in transformer networks. arXiv preprint arXiv:2410.17498.

**Questions:**

The authors explain KronSAE's improvements on reducing feature absorption in sec 4.2 as follows:
- Per "Smooth mAND activation", you state that "we introduce a differentiable AND gate [(mAND)] that prevents a broadly polysemantic primitive from entirely subsuming a more specific one." *It is not clear to me why this would be the case. Can you explain this rationale in greater detail?* (That is: I understand that, empirically, KronSAE demonstrably reduces absorption per the SAEBench tests -- which is impressive and significant -- and the second explanation of "Head-wise Cartesian decomposition" also seems intuitive; I just find the "Smooth mAND activation" explanation to be quite opaque. I think the second explanation is already sufficient to make your point, but if you can better clarify the first point as well, that would be helpful.)

One additional note for the authors is: I personally find the improvements in feature absorption (per sec 4.2) and interpretability/specificity/compositionality (per sec 5.3) to be much more significant contributions than what seem like comparatively modest efficiency gains. As such, I feel that motivating this work primarily from the perspective of efficiency (per the abstract and introduction) "undersells" the contribution.

---

> ### Author Response · Authors · 2025-11-23
> **Response to Review**
>
> Thank you for your thoughtful review and for highlighting the paper’s strengths - the novel application of Kronecker factorisation to SAEs, the joint focus on efficiency and interpretability, and the breadth of our experimental evaluation. Suggested papers are indeed related to our work and we will add them to the related work section with a brief discussion.
>
> Considering “Smooth mAND activation”, we agree that it is not as informative and clear as it might be, and the phrase “prevents a broadly polysemantic primitive from entirely subsuming a more specific one” is actually mistakenly written - it should be other way around. Thank you for pointing into this. We plan to restructure the section 4.2 with absorption by including new results obtained during rebuttal (with new baselines and in more consistent evaluation setting) and make presentation clearer (as in second explanation).
>
> Also thank you for your note! We appreciate your position and will slightly reformulate contribution part with less focus on efficiency and more on feature quality improvement in a future revision, as soon as we incorporate the other improvements so to not break the narrative in the paper.

---

### Official Review · Reviewer_cP5m · 2025-11-10

**Soundness:** 2
**Presentation:** 3
**Contribution:** 2
**Rating:** 6
**Confidence:** 2

**Summary:**

This paper introduces what the authors call KronSAE, an SAE architecture with an idea to address the scalability and interpretability challenges associated with large dictionary sizes in traditional SAEs. The primary optimisation involves factorising the latent representation using Kronecker product decomposition, which aims to reduce the computational and memory overhead of the dense encoder projection. This was identified as one of the key bottlenecks. The architecture also uses mAND, a differentiable activation that approximates the logical AND operation, encouraging compositional structure and improving latent quality.

**Strengths:**

- Applying Kronecker product decomposition to SAE seems new.  The tensor factorisation in model compression have been considered before (e.g. Edalati et al. 2021) but the specific application to SAE latent spaces with head-wise decomposition as far as the reviewer can see is novel.

- Aiming to address both computational efficiency (encoder bottleneck) as well as interpretability (compositional structure) is  a more ambitious task than usual optimisation-first approaches.

- The authors use a reasonably comprehensive exprimental setup involving 3 LLMs (Qwen, Pythia, Gemma), 3 dictionary sizes  (32k-131k), multiple token budgets.

- The paper is overall written with clarity and nicely presented figures, although some of the labels are too small (most figures).

**Weaknesses:**

- The paper mentions that Kronecker factorisation induces compositional/hierarchical features. But the mechanism seems to be underspecified and not rigorously justified/described

- The mAND activation (Eq. 3) is one of the important elements of the method but the paper seems to lack principled justification beyond empirical performance. Why square root? How closely does mAND approximate true binary AND? (Fig. 10)  How much the smoothness introduces unwanted activations (e.g. false positives in logical sense)?

- As considered by Bussmann at al. 2025, “Matryoshka SAEs” impose feature hierarchy by nested training. Is there a direct comparison in expriments? How similar/different is KronSAE’s hierarchy from Matryoshka?

**Questions:**

- Can you provide a more formal theoretical justification as for why Kronecker factorisation induces hierarchical/compositional features? What mathematical properties of the Kronecker product lead to semantic compositionality?

- Could you provide a more direct comparison to Matryoshka SAEs (Busmann et al, 2025), Switch SAEs (2025) and Gated SAEs (Rajamonoharan et al. 2024)? Since you briefly discuss these methods improve efficiency, but how does KronSAE compare quantitatively? Can you provide a more direct comparison under perhaps equal FLOP/paraemter budget?

---

> ### Author Response · Authors · 2025-11-21
> **Response to Weaknesses**
>
> Thank you for your thoughtful review and for highlighting the paper’s strengths: the novel application of Kronecker factorisation to SAE latent spaces, the joint focus on computational efficiency and interpretability, and the comprehensive experimental evaluation across multiple models and dictionary sizes. We appreciate your positive reading of the manuscript and will address weaknesses and questions below.
>
> W1. We will include more details about the motivation of architectural design in the main body in a revised version. We plan to cover how exactly Kronecker factorization induces hierarchy, as described in answer to Q1.
>
> W2. The square root is motivated by the following consideration: product $p*q$ (from the Kronecker product) might cause activation explosion - square root monotonously transform the result and preserves activation magnitude (as described in lines 111-112).
>
> Considering the approximation of true binary AND, from a logical point of view mAND produces nonzero activation only when both operands are nonzero - meaning that mAND is equal to AND in this case, with no false positives and false negatives. From a point of view of differentiable approximations, it is unknown to us how to compare the idealized logical AND with approximations introduced by us (mAND) and by [1] in a setting of sparse autoencoders due to lack of ground truth dataset of logical relationships in features and the lack of information about ground true features of LLM (every SAE finds its own feature set, and even the same SAE can find different features in different runs [2, 3, 4, 5]). See also our answer to reviewer GyXY, W1b. We will make the motivation clearer in a revision.
>
> W3. Matryoshka imposes different kind of hierarchy by dividing the dictionary into groups of G1, …, Gk latents where each is of different level of granularity. Namely, first G1 latents are the most broad and abstract, next G2 latents add more fine-grained semantics, lowering the level of abstraction, and so on. This type of structure imposed by specific loss function (increasing the level of granularity must decrease the reconstruction error, and the lowest level of G1 features must also maintain good enough reconstruction quality) and does not strictly demand some kind of conditional activation of features, while our approach imposes two-level AND-like hierarchy via architectural design (Kronecker product). We will add more discussion about Matryoshka and Switch SAE into the introduction and related work, since they are now also appear as baselines.
>
> Actually, KronSAE is orthogonal to Matryoshka SAE and could even be combined with it. We additionally ran experiments where with KronSAE versions of baseline architectures - please see the answer to Q2.

---

> ### Author Response · Authors · 2025-11-21
> **Response to Questions**
>
> Q1. Consider base and extension pre-latent vectors p and q to which ReLU was applied. Each $p_i$ is a ith base pre-latent, and $q_i$ is ith extension pre-latent. Kronecker product $p \otimes q$ creates the vector $(p_1 * q_1, …, p_1 * q_m, p_2 * q_1, …, p_n * q_m)$ of the activations of post-latents. Then there is two situations:
>
> 1. Post-latent is active → both pre-latents activations are positive.
> 2. Post-latent is inactive → at least one of pre-latent activations is zero.
>
> Therefore, post-latent is active only when both pre-latents are active. Fix some post-latent and its corresponding pre-latents, and suppose that $\mathcal{P}, \mathcal{Q}, \mathcal{F}$ are the sets of input vectors (passed to SAE) on which base pre-latent, extension pre-latent and post-latent are active. Then it is true that $\mathcal{F} = \mathcal{P} \cap \mathcal{Q}$, meaning that pre-latents must be broader and polysemous to encode multiple semantics of emitted post-latents. We validate this behaviour qualitatively in section 5.3, although apparently other types of interactions are also presented (see lines 414 - 424).
>
> Q2. We have ran additional experiments with Matryoshka and Switch SAEs as baselines;  we already have an ablation experiment with the JumpReLU activation in appendix A.4 which is a natural simplification of the Gated SAEs and serves as a stronger baseline [6]. We perform this experiments on both Qwen-2.5 1.5B and Gemma-2 2B models under the iso-FLOPS setting as described in our paper.
>
> The following results show the performance of the baselines and their KronSAE variants on 65k dictionary size and with 500M tokens as reference budget, all SAEs aligned with TopK SAE FLOPS:
>
> https://postimg.cc/0MBrnDM1
>
> The next results on feature absorption suggest that applying KronSAE encoder improves absorption scores, especially combined with Matryoshka dictionary learning:
>
> https://postimg.cc/3yZvXkF8
>
> The following results are for interpretability scores:
>
> 32k dictionary: https://postimg.cc/GBB6f436
>
> 65k dictionary: https://postimg.cc/WDJyL9xx
>
> We will include all these results in the paper with detailed description of hyperparameters.
>
> We note that KronSAE is more than another SAE variant: its head-wise Kronecker-factorised encoder and mAND operator are architectural primitives that can be applied to any SAE with a dense encoder (not TopK only). All these approaches are orthogonal to each other: one could build the Switch SAE with Kronecker-factorized experts (as in KronSAE) trained with Matryoshka-like loss, combined with JumpReLU activation function.
>
> ---
>
> Thank you again for your review. Please let us know if you have further questions, suggestions or concerns to address.
>
> ---
>
> [1] Logical Activation Functions: Logit-space equivalents of Probabilistic Boolean Operators, Lowe et al., 2022
>
> [2] Sparse Autoencoders Do Not Find Canonical Units of Analysis, Leask et al., 2025
>
> [3] Sparse Autoencoders Trained on the Same Data Learn Different Features, G. Paulo, N. Belrose, 2025
>
> [4] Archetypal SAE: Adaptive and Stable Dictionary Learning for Concept
> Extraction in Large Vision Models, Fel et al., 2025
>
> [5] Position: Mechanistic Interpretability Should Prioritize Feature Consistency in SAEs, Song et al., 2025
>
> [6] Jumping Ahead: Improving Reconstruction Fidelity with JumpReLU Sparse Autoencoders, Rajamanoharan et al., 2024

---

### Author Response · Authors · 2025-12-03
**General Response**

Dear Reviewers and Area Chair,

We sincerely thank all reviewers (**cP5m**, **Hh9i**, **eE5v, dNqs, GyXY**) for their time and detailed and constructive feedback. We are encouraged that reviewers highlighted (a) novelty and usability of applying Kronecker factorization to Sparse Autoencoders  (**cP5m, eE5v, Hh9i**), (b) computational efficiency of the method (**cP5m, Hh9i, GyXY**), (c) rich variety and quality of experiments, including multiple LLMs, dictionary sizes and token budgets (**cP5m, Hh9i, eE5v, dNqs**) and (d) clarity of presentation (**Hh9i, cP5m, dNqs**).

We have conducted additional experiments and clarified many parts of the paper in response to reviewers comments and suggestions, and below we summarize the work done during rebuttal and improvements incorporated in a revised version of the paper that resolve the reviewers concerns.

**Matryoshka and Switch baselines experiments (Reviewers cP5m, dNqs, GyXY).** To compare our architecture and show possibility to apply KronSAE to existing SAEs we include Matryoshka and its KronSAE version in all main evaluations across two models Gemma-2 2B and Qwen-2.5 1.5B with dictionary sizes 65k and 32k with sparsity levels $\ell_0 = \{16, 32, 64, 128\}$ (*Figures 1, 3, 4, 9, 10, 11, 20 and 21*). Additionally, we include Switch SAE and its KronSAE version in majority of main evaluations (*Figures 3, 10, 20 and 21*). In the Appendix A we include details on how we train these baselines to ensure reproducibility.

**Sparsity experiments (Reviewer dNqs)**. To address concern about generalization of our architecture to different sparsity levels we evaluate it across $\ell_0 = \{16, 32, 64, 128\}$ on 65k and 32k dictionary sizes (*Figures 3, 4, 10, 11, 20 and 21*). KronSAE maintain same reconstruction quality while using less parameters and scales comparably to TopK.

**Toy Model of Correlation (Reviewer eE5v).** To address reviewer **eE5v**’s question about whether learned heads depend on the ground-truth correlation structure, we decided to refactor this experiment and make it more clear and rigorous. In *Section 5.1 and Appendix B of revised paper* we change our feature matching methodology and isolate random permutation factor of SAE features. We show both qualitatively and visually that our parameterization highly dependent on the given correlation pattern.

**Consistency in presentation (Reviewer dNqs).** We have substantially improved the consistency of the reporting following the reviewer comment about its necessity, and now all main evaluations in the paper are now conducted on two models, different sparsity levels and several dictionary sizes (Figures 1, 3, 4, 9, 10, 11, 20 and 21). We also added Matryoshka baseline on main figures (Figures 1 and 9 in revised version) with three dictionary sizes.

**Theoretical justification of architecture (Reviewers cP5m, Hh9i, eE5v, GyXY).** To make the motivation behind KronSAE clearer, we have included additional explanation of AND-like behavior in the *Section 3* as a short note and in *Appendix C of revised paper* as expanded discussion. In these clarifications we strengthened the connection between mAND, Kronecker product and the logical AND, explaining how our architecture implements an logical AND-like activation and hierarchy.

**Matryoshka and KronSAE hierarchy (Reviewers cP5m, GyXY)**. We expand our explanations of KronSAE mechanisms in *Appendix C* for Matryoshka hierarchy discussion compared to our architecture and AND-like mechanism explanation.

**Tensor product discussion (Reviewers Hh9i, eE5v)**. We add short discussion of Tensor Product Representations and its relation to our mAND idea in *Related Work* section.

**Structured matrices (Reviewer eE5v).** We include reference to the structured matrices in *Related Work* section.

**Interpretability results and tradeoff (Reviewer dNqs).** To enhance the reporting of interpretability results and KronSAE trade-offs, we conducted additional experiment with several $m, n$ configurations and include the results the *Figure 22* and the discussion of trade-off in *Appendix E of revised paper*.

**Larger budgets (Reviewer dNqs).** To show KronSAE scaling on budgets larger than those studied in the paper, we train KronSAE and TopK SAE on iso-FLOPs setup with 2B reference tokens. Results are shown in *Figure 1* of revised version.

**Figures clarity.** We have improved the visibility of the text and legends on Figures and their placement to make the presentation clearer.

**Diagram of the method (Reviewer eE5v).** *Figure 23* now includes improved diagram of KronSAE method.

---

> ### Author Response · Authors · 2025-12-03
> **General Response**
>
> Taken together, these additions substantially strengthen the paper and address the reviewers’ concerns regarding baselines, sparsity regimes, theoretical motivation, interpretability reporting, and experimental consistency. The revised manuscript now includes comprehensive comparisons (TopK, Matryoshka, Switch and their Kron variants), expanded theoretical justification of the mAND mechanism and Kronecker architecture, deeper interpretability and absorption analyses across hyperparameters, and clearer, more rigorous toy-model evaluations. We are grateful to the reviewers for their dedication and contribution to our work and we respectfully believe that the updated paper provides a clear, well-supported contribution to the field of mechanistic interpretability, and we hope the revisions satisfactorily resolve the outstanding concerns raised during review.

---

### Meta-Review · Area_Chair_idnP · 2026-01-04

**Summary:**

The paper proposes KronSAE, a sparse autoencoder architecture utilizing Kronecker factorization to improve computational efficiency and interpretability by enforcing a hierarchical structure on latent representations. While reviewers generally agreed on the novelty of applying Kronecker factorization to SAEs and the breadth of the experimental setup involving multiple LLMs, there were significant concerns regarding the experimental rigor and the validity of the interpretability claims.

The reviewers' concerns primarily focused on:
1. Experimental Baselines: Initial lack of comparison to relevant baselines like Matryoshka, Switch, or Gated SAEs.
2. Trade-offs: A critical concern that the paper obscured a trade-off between reconstruction quality and interpretability/sparsity. Specifically, settings that achieved competitive reconstruction ($m=1$) behaved like standard SAEs, while settings that offered interpretability benefits ($m>1$) incurred reconstruction penalties.
3. Validity of Structural Priors: Skepticism regarding whether the architecture genuinely uncovers ground-truth hierarchy or merely imposes it due to architectural constraints.
4. Missing Citations: Lack of reference to Tensor Product Representations.

**Reviewer Concerns:**

Addressed Concerns:


1. Baselines: The authors conducted extensive additional experiments during the rebuttal to compare against Matryoshka and Switch SAEs.



2. Clarifications: The authors provided clarifications regarding the "mAND" activation function and provided diagrams to explain the method better.



3. Citations: The authors agreed to include missing references to Tensor Product Representations.

Outstanding Concerns:
1. The Reconstruction/Interpretability Trade-off: Reviewer dNqs remained concerned that the paper does not adequately transparently report the trade-off between the hyperparameters ($m$) that yield good reconstruction versus those that yield the claimed interpretability benefits. The authors admitted this trade-off exists, but the reviewer's concern suggests the current manuscript "hides" this dynamic.

2. Motivation for Structural Prior: Reviewer GyXY's concern regarding the theoretical motivation for why this specific 2-level hierarchy is the correct prior for LLM features remains largely philosophical and not fully resolved by the empirical data.

3. Validity of Correlation Experiments: Reviewer eE5v flagged a "blocker" concern that the architecture might be "finding" correlated structures simply because the architecture enforces them, regardless of the underlying ground truth. While the authors provided new RV coefficient analysis, the fundamental concern about circularity in the evaluation of the architectural prior persists.

**Reviewer Scores:**

1. Reviewer cP5m (Score: 6): Likely to remain at 6 or potentially drop to 5. While they appreciated the rebuttal, they explicitly stated they "would not mind if paper is rejected", suggesting weak support.

2. Reviewer Hh9i (Score: 10): Likely to remain high (8-10). This reviewer was very enthusiastic and felt the efficiency gains were undersold. They might lower slightly if they engaged deeply with dNqs's point about the hidden trade-offs.

3. Reviewer eE5v (Score: 6): The reviewer identified the correlation experiment issue as a "blocker" and questioned whether the architecture simply "works around" structural constraints rather than validating them.

4. Reviewer dNqs (Score: 2): Likely to remain low. The rebuttal confirmed their suspicion about the trade-off.

5. Reviewer GyXY (Score: 2): Likely to remain low. The core issue of motivating the specific hierarchical prior is difficult to resolve with just additional experimental plots.

---

### Decision · Program_Chairs · 2026-01-26

Reject